# MOGIC: Metadata-infused Oracle Guidance for Improved Extreme Classification

Suchith Chidananda Prabhu [* 1]  Bhavyajeet Singh [* 2]
Anshul Mittal [3]  Siddarth Asokan [2]  Shikhar Mohan [3]  Deepak Saini [4]  Yashoteja Prabhu [2]
Lakshya Kumar [3]  Jian Jiao [4]  Amit S [3]  Niket Tandon [2]  Manish Gupta [3]  Sumeet Agarwal [1]  Manik Varma [2]

## Abstract

Retrieval-augmented classification and generation models benefit from *early-stage fusion* of high-quality text-based metadata, often called memory, but face high latency and noise sensitivity. In extreme classification (XC), where low latency is crucial, existing methods use *late-stage fusion* for efficiency and robustness. To enhance accuracy while maintaining low latency, we propose MOGIC, a novel approach to metadata-infused oracle guidance for XC. We train an early-fusion oracle classifier with access to both query-side and label-side ground-truth metadata in textual form and subsequently use it to guide existing memory-based XC disciple models via regularization. The MOGIC algorithm improves precision@1 and propensity-scored precision@1 of XC disciple models by 1–2% on six standard datasets, at no additional inference-time cost. We show that MOGIC can be used in a plug-and-play manner to enhance memory-free XC models such as NGAME or DEXA. Lastly, we demonstrate the robustness of the MOGIC algorithm to missing and noisy metadata. The code is publicly available at `https://github.com/suchith720/mogic`.

## 1. Introduction

Context enrichment is the process of incorporating metadata from external sources (referred to as **memory** for a model, comprising *memory items*) to improve the overall performance on a task. While such methods have gained popularity in generative modeling in the form of retrieval-augmented generation (RAG) (Lewis et al., 2020; Gao et al., 2024; Fan et al., 2024), context enrichment also benefits classification tasks (Guu et al., 2020b; Guo et al., 2023; Mohan et al., 2024; Long et al., 2022). In this paper, we focus on using memory to improve classification, and in particular scenarios wherein the label space is extremely large (typically of the order of millions), *i.e.,* eXtreme Classification (**XC**) (Mohan et al., 2024).

Real-world XC tasks often involve short-text queries, such as in matching queries to advertiser-bidded keywords (for sponsored search ads (Dahiya et al., 2021; Jain et al., 2016; Prabhu et al., 2018b)), queries to product titles (for product recommendations (Dahiya et al., 2021; Medini et al., 2019; Mittal et al., 2021)), or queries to webpage titles (for tagging (Babbar & Schölkopf, 2017; Chang et al., 2020; You et al., 2019)). These XC tasks, characterized by an extremely large label space, often suffer from data sparsity, having very few training examples per label on average. In such XC tasks, auxiliary metadata can offer relevant diverse information, which can be leveraged to obtain a better understanding of the input queries, thereby making accurate predictions. For example, on the query-to-ad-keyword prediction task, the query-side metadata can be obtained by mining organic search webpage titles clicked in response to the query on the search engine, while on the Wikipedia categories prediction task, other Wikipedia article titles connected to the original page via hyperlinks could serve as the metadata. For a detailed overview of the preliminaries, and definitions of key terms, please refer to Appendix A.

Memory-based augmentation in XC remains a challenging problem due to the need for accurate and fast memory access. The low latency requirements are imposed by real-world applications. Ideal/golden memory item linkages are typically available during training, referred to as *ground-truth metadata*. However, these golden linkages are seldom available during real-world deployment, particularly in real-world scenarios, where 80-85% of the traffic is in the *cold-start* setting, as reported in (Mittal et al., 2025). Our work demonstrates that methods that rely entirely on high-quality ground-truth linkages of metadata are susceptible to

---

[*]Equal contribution  [1]Yardi School of Artificial Intelligence, IIT Delhi, India [2]Microsoft Research, India [3]Mircosoft, India [4]Mircosoft, USA. Correspondence to: Suchith Chidananda Prabhu <suchith720@gmail.com>.

*Proceedings of the 42^{nd} International Conference on Machine Learning*, Vancouver, Canada. PMLR 267, 2025. Copyright 2025 by the author(s).

Table 1: A comparison of predictions from MOGIC (OAK), OAK, MOGIC (NGAME), and NGAME models, and the ground-truth and oracle predictions, on the Wikipedia *See Also* prediction task. Legend: Black indicates ground truth, Red indicates incorrect predictions, Green indicates correct predictions, and Blue indicates missing labels (as per a human judge).

| QUERY | Grass court | Tangbe |
|---|---|---|
| GROUND TRUTH LABELS | Clay court, Carpet court, Hardcourt | Mustang District, Kali Gandaki Gorge, Kali Gandaki River, Upper Mustang, Gandaki River |
| GROUND TRUTH QUERY METADATA | Tennis terminology, Sports rules and regulations, Tennis court surfaces | Populated places in Mustang District |
| PREDICTED QUERY METADATA | Courts by type, Landforms, Grasslands | Populated places in Cameroon, Communes of Cameroon, Township-level divisions of Hebei |
| ORACLE PREDICTIONS | Clay court, Carpet court, Hardcourt, Plexicushion, DecoTurf | Mustang District, Kali Gandaki Gorge, Kali Gandaki River, Upper Mustang, Mustang Caves |
| OAK PREDICTIONS | Fernie Ghostriders, Garland, Texas, List of Nevada state prisons, Ronald Reagan Boyhood Home, West End (Richmond, Virginia) | Desalpur, Second Franco-Dahomean War, Vladivostok, Kitenge, List of currently erupting volcanoes |
| MOGIC (OAK) PREDICTIONS | Clay court, Carpet court, Hardcourt, Video arcade, U.S. Men's Clay Court Championships | Mustang District, Kali Gandaki Gorge, Kali Gandaki River, Upper Mustang, Gandaki River |
| NGAME PREDICTIONS | National Register of Historic Places listings in Cumberland County, North Carolina, Shangri La (Doris Duke), Vauxhall, National Register of Historic Places listings in Perry County, Alabama | Five kings of Wa, Piteraq, Wemale, Oyo Empire, List of lighthouses in Togo |
| MOGIC (NGAME) PREDICTIONS | Clay court, Carpet court, Hardcourt, Stadium, Riding hall | Mustang District, Upper Mustang, Mustang Caves, List of municipalities in Andhra Pradesh, Muktinath |

noise and fail to make accurate predictions, even when these linkages are slightly perturbed (see Table 5). To address the cold-start challenge, prior work focused on predicting metadata during inference as an auxiliary task. This approach typically involves training a separate model on ground-truth metadata to solve this auxiliary task. However, several challenges persist, and can be characterized into: (a) *Sensitivity to retrieved metadata*: Low quality retrievals from memory lead to noisy augmentations to the query, degrading task performance. For example, Cuconasu et al. (2024); Yoran et al. (2024) and Yu et al. (2024) showed that, for text-based early-fusion models, robustness to retrieved memory item quality is critical for performance. (b) *Relationship between latency and the form of metadata infusion*: Textual metadata offers superior interpretability over embedding-based metadata, but can lead to high inference-time fusion latency as it causes an increase in the sequence length of the input to the transformer.

State-of-the-art retrieval augmentation approaches struggle to balance accuracy and efficiency in XC. Recent works such as OAK (Mohan et al., 2024) and PINA (Chien et al., 2023) have shown that using metadata can improve online classification performance. However, each approach has its limitations. For example, OAK's late-stage embedding-based fusion method improves generalization and reduces inference latency when handling noisy predicted metadata (achieving precision@1 of 33.71 compared to 28.49 for early-fusion on LF-WikiSeeAlsoTitles-320K dataset). However, even when provided with the ground-truth metadata, the OAK model produces suboptimal representations com-

pared to early-fusion models (achieving precision@1 of 38.92 compared to 47.63 for early-fusion). This highlights a key trade-off: while early-fusion models may struggle with noisy metadata, they demonstrate superior performance when given clean, high-quality metadata. In this work, we aim to leverage this insight, using early-fusion models to improve the performance of memory-based XC models. Furthermore, this XC setting uniquely allows for incorporating label-side metadata within the memory framework, a feature that has not been fully utilized in existing approaches.

In this paper, we design a novel memory-based approach for XC that utilizes both query and label-side metadata and maintains low inference latency, by employing a combination of early-fusion of text-based metadata and late-fusion of memory items. Our approach involves two phases of training: (a) *Oracle* training, and (b) *Oracle-guided disciple* training. We call this two-stage metadata-infused oracle guidance framework for improved extreme classification as MOGIC. In the first phase, we train an early-fusion *Oracle* classifier which is provided access to both query-side and label-side ground-truth metadata in textual form. In the second phase, the oracle is used to guide the training of any existing memory-based XC disciple model (such as OAK or NGAME (Dahiya et al., 2023a)), by means of a novel regularization loss. Table 1 shows two examples of queries with ground truth labels; predicted and ground truth memory items; the oracle predictions, and predictions from OAK, NGAME, and their MOGIC counterparts on the Wikipedia *related articles* prediction task. We observe that for the query "Grass Court", "Courts by type" is a relevant memory

item. However, some predicted metadata (e.g., "Landforms" and "Grasslands") mislead OAK, leading to incorrect predictions about geographical places. NGAME also exhibits poor performance in this scenario. In contrast, MOGIC, trained with our novel regularization, effectively retains the original intent of the query. This is because MOGIC leverages the best of both worlds – it maintains low inference latency and effectively handles noisy predicted metadata through late-fusion, while simultaneously learning to mimic the superior embeddings of the early-fusion oracle through our novel regularization loss. This combined approach mitigates drawbacks of both late- and early-fusion methods, resulting in improved performance. Appendix H contains a detailed case study on how metadata and oracle-guided training help MOGIC rectify errors from OAK.

In summary, our contributions are as follows:

- We propose **MOGIC**, a Metadata-infused Oracle Guidance framework for Improved Extreme Classification, that maintains real-world inference latency, while achieving 1–2% improvement over state-of-the-art XC models.

- We develop a novel regularization loss that leverages the oracle model to guide the training of a memory-based disciple. This approach enables the disciple to demonstrate performance gains while maintaining low inference latency, resulting in a 50% reduction in computational costs compared to the oracle.

- By means of extensive experimentation on six popular XC datasets, we show that (a) MOGIC improves accuracy significantly in terms of standard metrics such as precision, NDCG and propensity-scored precision atop both memory-based models such as OAK as well as memory-free models such as DEXA and NGAME; (b) MOGIC is robust to missing and noisy metadata compared to the oracle, and (c) MOGIC (OAK) achieves state-of-the-art metrics across all datasets.

## 2. Related Work

**Extreme Classification**: XC is a crucial component in ranking and recommendation systems (You et al., 2019; Guo et al., 2019; Dahiya et al., 2021; Mittal et al., 2021; Saini et al., 2021; Gupta et al., 2023; Mohan et al., 2024). XC approaches learn a classifier associated with each label, treating each of the labels as classes in the multi-label setting, with features obtained either via classical approaches such as bag-of-words (Babbar & Schölkopf, 2017; Prabhu et al., 2018b), decision trees (Prabhu et al., 2018b) or via deep-learning techniques that leverage either pre-trained (Jain et al., 2019) or learned (You et al., 2019; Jiang et al., 2021; Dahiya et al., 2023a) models. The closest approach to ours is that of OAK (Mohan et al., 2024), wherein an XC classi-

fier, such as an NGAME (Dahiya et al., 2023a) encoder, is used to retrieve the metadata, and a single transformer attention layer is used to fuse the representations of the query and the retrieved metadata. The proposed MOGIC algorithm leverages a text-based early-fusion model to improve the representations of the memory items in OAK. Along another direction, models such as DEXA (Dahiya et al., 2023b) aggregate information from the neighborhood of the encoder representations to form the context. Consequently, MOGIC can also be applied atop XC models such as NGAME and DEXA to improve performance (cf. Table 6).

**Retrieval-augmented Generation (RAG)**: The RAG paradigm has become the *defacto* approach for incorporating metadata for context enrichment in generative models, with the application typically being that of question answering. Prior to RAG, models such as REALM (Guu et al., 2020a) leveraged external knowledge sources to improve the accuracy of the transformer encoders using a retriever that selected relevant documents or passages from the memory, while an encoder fused the input text and memory items to compute an enriched embedding. RAG-based approaches (Lewis et al., 2020; Akyurek et al., 2023; Zhang et al., 2023; Radhakrishnan et al., 2024; Muennighoff et al., 2024) combine pre-trained parametric and non-parametric memory for language generation. In RAG settings, the memory, typically text-based, is infused with the query at the input (Yang et al., 2018; Karpukhin et al., 2020; Qu et al., 2021; Lan et al., 2023; Lála et al., 2023; Yan et al., 2024). Other approaches incorporate task-specific memory, such as tabular data (Zha et al., 2023; Luo et al., 2023) or knowledge graphs (Gaur et al., 2022; He et al., 2024). We observe that retrieval-augmented models benefit from early-fusion, and high-quality metadata, but suffer from high inference latency and poor robustness to noise. MOGIC makes a novel contribution by introducing the textual early-fusion of metadata into XC models while respecting latency constraints.

**Guided Representation Learning**: Transferring capabilities via context-following from large language models (LLMs) to smaller ones has been widely studied (Kim & Rush, 2016; Gupta & Agrawal, 2022; Xu et al., 2024). In the generative setting, models such as Alpaca (Taori et al., 2023), Vicuna (Chiang et al., 2023), Self-instruct (Wang et al., 2023), etc., have been shown to use supervised instruction-following fine-tuning to improve generation, where the tuning data is generated using LLMs. On the LLM-based classification task, AugGPT (Dai et al., 2025) employs a teacher LLM to rephrase input sentence to improve general and clinical-domain classification performance. Various other works (Gilardi et al., 2023; He et al., 2023; Gao et al., 2023; Chenglin et al., 2024; Li et al., 2023) have considered guidance for LLM-based classification in the context of annotation generation, data clustering and curation, etc., but do not target the XC setting. The Ora-

cle guidance framework in MOGIC can be viewed as an instantiation of guided representation learning, wherein the text-based early-fusion Oracle provides supervision for a downstream model such as OAK. MOGIC is therefore orthogonal to these existing guidance-based approaches.

## 3. The MOGIC Approach

Before we describe the MOGIC algorithm, we first introduce the notation used in this paper.

**Notation:** Consider the task of query-to-label subset prediction, as common in the XC setting. Let $L$ denote the total number of labels present, and $Q$, the total number of queries. Let $X_q$, $Z_l$ be the textual descriptions of the query and the label, indexed by $q$ and $l$, respectively. For each query $X_q$, its ground-truth label vector is $\mathbf{y}_q \in \{-1, +1\}^L$, where $[\mathbf{y}_q]_l = y_{ql} = +1$ if label $l$ is relevant to query $q$ and $y_{ql} = -1$ otherwise. Let $A_k$ be the textual descriptions of the auxiliary memory item (indexed by $k$), and $M$ denote the total number of memory items. For each query $X_q$ or label $Z_l$, the vector representation of it's ground truth memory-items is denoted by $\mathbf{a}_{q/l}$ with $\mathbf{a} \in \{-1, +1\}^M$, where $a_k = +1$ if memory item $k$ is relevant and vice-versa. We assume that the label and the memory-item sets remain unchanged between training and testing.

In summary, $\mathcal{X} \overset{\text{def}}{=} \{X_q\}_{q=1}^Q \cup \{Z_l\}_{l=1}^L \cup \{A_k\}_{k=1}^M$ denote all the textual information, comprising $Q$ labeled queries, $L$ labels, and $M$ memory items. The dataset is then denoted by $\mathfrak{D} \overset{\text{def}}{=} \{\{X_q, \mathbf{y}_q, \mathbf{a}_q\}_{q=1}^Q, \{Z_l, \mathbf{a}_l\}_{l=1}^L, \{A_k\}_{k=1}^M\}$. Within this setting, the XC problem is one of predicting relevant set of labels $\tilde{\mathbf{y}}_q$ associated with each query $X_q$, while leveraging memory items. We now present the MOGIC framework.

### 3.1. The MOGIC Framework

MOGIC comprises four main components (a) The base XC model (disciple), either memory-based, or memory-free; (b) The oracle $\mathcal{O}$ for guidance. (c) The XC-task-specific loss function; and (d) The guidance loss function. In this paper, we primarily focus on memory-based XC models, and in particular, OAK. Please refer to Figure 1 for an overall understanding of the integration of OAK into the MOGIC framework. Appendices B and C provide additional details. The four blocks are described below.

1. **Disciple** ($\mathcal{D}$): The disciple (in this case, OAK) is a trainable XC architecture with parameters $\theta_\mathcal{D}$ such that this encoder model takes in a query or label and outputs a $d$-dimensional embedding $\mathcal{E}_{\theta_\mathcal{D}} : \mathcal{X} \to \mathcal{S}^{d-1}$ that lies on the unit sphere $\mathcal{S}^{d-1}$.

2. **Oracle** ($\mathcal{O}$): The oracle is an encoder model (either a large language model (LLM) or a small language model (SLM)) with parameters $\theta_\mathcal{O}$ and is used to guide

the disciple in the second phase of the MOGIC framework. Typically the oracle is a computationally expensive, but highly accurate embedding-based model. The oracle accepts as input, queries, labels and their associated metadata, and generates high-quality representations, while being trained/finetuned for the XC task. These embeddings are subsequently used to guide the disciple via a guidance loss. In MOGIC, we consider DistilBERT (Sanh et al., 2019), LLaMA-2 (Touvron et al., 2023) and Phi-2 (Javaheripi et al., 2023) finetuned on the XC task as oracles.

3. **Task-specific Loss Function**: The loss, denoted by $\mathcal{L}_{\texttt{Disciple}}$, is associated with the task for which the disciple is being trained. For instance, in OAK, $\mathcal{L}_{\texttt{Disciple}}$ is a triplet margin loss (Mohan et al., 2024).

4. **Guidance Loss Function**: The guidance is provided by the oracle to the disciple using two loss functions: $\mathcal{L}_{\texttt{Alignment}}$ and $\mathcal{L}_{\texttt{Matching}}$ (cf. Section 3.3).

MOGIC is a highly modular framework which can accommodate different choices of the oracle model, the disciple model and its task-specific loss function. The following section discusses the training of the oracle and the guidance loss employed to regularize the disciple.

### 3.2. MOGIC Phase 1: Oracle Training

To train a highly accurate XC oracle, three components are critical: (a) The task-specific loss function; (b) Supervised training data, and (c) Auxiliary metadata, which can enhance the quality of label and query. Prior work has predominantly focused on designing effective task-specific loss functions (Dahiya et al., 2023a; Gupta et al., 2023; Kharbanda et al., 2023). We leverage the standard triplet loss with in-batch negative sampling for training our oracle model. The oracle is trained by using early-fusion, wherein the query, label, and their associated metadata are provided at the input layer via simple text concatenation (with appropriate delimiter tokens). The input is then projected onto an embedding space $\mathcal{R}^d$ using an encoder $\mathcal{E}_{\theta_\mathcal{O}}$. For example, given a query $q$ with textual description $X_q$, the input to the oracle is the query text concatenated with its associated $m$ memory items, i.e., $\tilde{X}_q = X_q || A_{q_1} || \dots || A_{q_m}$, where $||$ denotes the delimiter token, and the corresponding embedding is given as $\mathbf{x}_q^* = \mathcal{E}_{\theta_\mathcal{O}}(\tilde{X}_q)$. On the label side, the metadata-enriched label representation is computed as $\mathbf{z}_l^* = \mathcal{E}_{\theta_\mathcal{O}}(\tilde{Z}_l)$, where $\tilde{Z}_l$ is defined in a manner similar to $\tilde{X}_q$. The optimization objective for the oracle is:

$$\theta_\mathcal{O} = \arg\min_\theta \mathcal{L}\Big(\{\mathbf{x}_q^*\}_{q \in Q}, \{\mathbf{z}_l^*\}_{l \in L}, \{y_{ql}\}_{q \in Q, l \in L}\Big)$$

where $\mathcal{L}$ can be any discriminative loss function that pushes relevant labels closer to and irrelevant labels farther away from the query in their joint embedding space. Based on

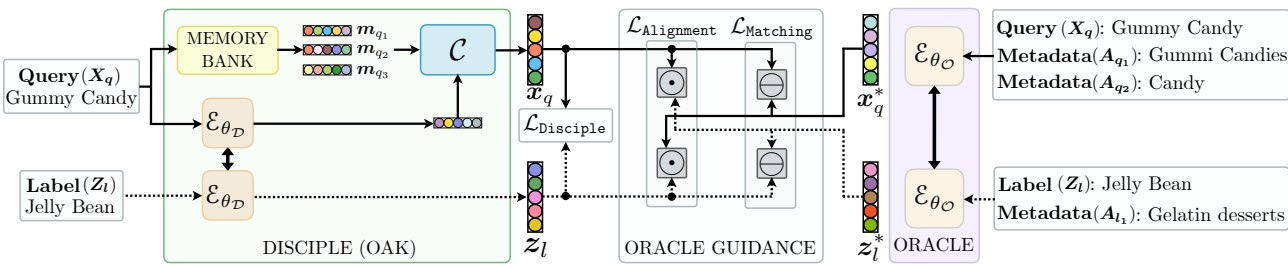

Figure 1: The MOGIC training framework can be used with any XC approach. In this figure, MOGIC is used over the OAK disciple and its task-specific loss. An oracle's (LLaMA-2, Phi-2 or DistilBERT) embeddings are used to regularize the representation of OAK using the guidance loss. The green box represents the OAK disciple architecture, comprising encoder ($\mathcal{E}_{\theta_\mathcal{D}}$), memory bank and the combiner ($\mathcal{C}$). The oracle comprises the encoder ($\mathcal{E}_{\theta_\mathcal{O}}$). The MOGIC approach involves training the disciple model on three loss: (a) The task loss $\mathcal{L}_{\texttt{Disciple}}$; (b) The oracle-disciple ranking alignment loss $\mathcal{L}_{\texttt{Alignment}}$, and (c) The oracle-disciple embedding matching loss $\mathcal{L}_{\texttt{Matching}}$.

empirical success, we use a triplet-loss-based optimization, which has demonstrated superior performance of dual-encoder-based models:

$$\mathcal{L}_{\texttt{Triplet}}\Big(\{\mathbf{x}_q\}, \{\mathbf{z}_l\}, \{y_{ql}\}\Big) = \sum_{\substack{p\,:\,y_{qp}=+1 \\ n\,:\,y_{qn}=-1}} [\mathbf{x}_q^\top \mathbf{z}_n - \mathbf{x}_q^\top \mathbf{z}_p + \gamma]_+,$$

where $\mathbf{z}_p$ and $\mathbf{z}_n$ are the embeddings of the positive and negative label, respectively, $\mathbf{x}_q$ is the query embedding, and $\gamma$ is the margin. We omit superscripts and indexing subscripts in the equation above for notational simplicity.

For the LLaMA-2 and Phi-2 SLMs, we perform LoRA fine-tuning for the specific XC task using the corresponding supervised training data and simple prompting (cf. Appendix D). For DistilBERT, we perform standard finetuning.

### 3.3. MOGIC Phase 2: Oracle-guided Disciple Training

The oracle typically demonstrates high accuracy on the downstream XC task due to its larger size and access to privileged information (ground truth *textual* metadata) not accessible to the disciple. However, they are impractical to deploy in any real-world application due to one or more of the following limitations: (a) High computational cost during deployment; (b) Slow inference time, or (c) A lack of robustness to noisy metadata. Therefore, we use guidance, in the form of embeddings from the oracle, to regularize a chosen disciple model. The inspiration for training a disciple model using an oracle is drawn from from knowledge distillation, however, the proposed framework has several key differences compared to traditional distillation, as discussed in Appendix C.1. While we present experimental results using multiple disciples in Section 4, here, we base our discussion around the OAK disciple as it has been shown to be the state-of-the-art for XC tasks.

Disciple training comprises two key components: (a) An embedding generator which provides embeddings $\mathbf{x}_q$ and $\mathbf{z}_l$, associated with a given query $X_q$ and label $Z_l$, respectively

and (b) A task-specific loss function over which the disciple is trained. In MOGIC, we propose two additional loss terms, namely the Alignment loss and the Matching loss, to provide oracle-guidance to the disciple for learning superior embeddings. We describe these two loss components below.

1. **Alignment**: The Alignment loss focuses on aligning the rankings of the oracle and the disciple. To enforce this, MOGIC introduces a triplet margin loss between the oracle's query and the disciple's label embeddings, and vice versa, as follows.

$$\begin{aligned} \mathcal{L}_{\texttt{Alignment}} = \ &\mathcal{L}_{\texttt{Triplet}}\Big(\{\mathbf{x}_q\}, \{\mathbf{z}_l^*\}, \{y_{ql}\}\Big) \\ &+ \mathcal{L}_{\texttt{Triplet}}\Big(\{\mathbf{x}_q^*\}, \{\mathbf{z}_l\}, \{y_{ql}\}\Big) \end{aligned} \quad (1)$$

Recall that the asterisk in the superscript indicates an embedding from the oracle model.

2. **Matching**: The Matching loss focuses on ensuring that the disciple mimics the oracle's embeddings. To enforce this in MOGIC, we introduce an L2 loss between the oracle query embeddings and the disciple query embeddings (and similarly for labels) as follows:

$$\mathcal{L}_{\texttt{Matching}} = \sum_{q \in Q} \big\| \mathbf{x}_q - \mathbf{x}_q^* \big\|_2 + \sum_{l \in L} \big\| \mathbf{z}_l - \mathbf{z}_l^* \big\|_2 \quad (2)$$

where $\mathbf{x}_q = \mathcal{E}_{\theta_\mathcal{D}}(X_q)$ and $\mathbf{z}_l = \mathcal{E}_{\theta_\mathcal{D}}(Z_l)$ are the query and label embeddings of the disciple, respectively. Finally, MOGIC combines the aforementioned two losses with the disciple's task-specific loss function as follows:

$$\mathcal{L}_{\texttt{MOGIC}} = \mathcal{L}_{\texttt{Disciple}} + \alpha \cdot \mathcal{L}_{\texttt{Alignment}} + \beta \cdot \mathcal{L}_{\texttt{Matching}} \quad (3)$$

where $\alpha, \beta$ are tunable hyper-parameters and set to $1.0$ and $0.1$ respectively in our experiments. Appendix G.1 provides a detailed sensitivity analysis over various values of $\alpha$ and $\beta$. During the guidance training phase, only the disciple is trained, while the oracle remains frozen.

## 3.4. Theoretical Justification of Oracle-Guided Losses

Oracle-guided training, where a frozen oracle guides the training of a disciple model through the `Alignment` and `Matching` losses, enhances the disciple's accuracy and robustness, by enabling implicit knowledge transfer and improving convergence with fewer training samples. We theoretically analyze the impact of minimizing these losses on the disciple's accuracy and measure the sample complexity under the following simplifying assumptions: (a) A Lipschitz-continuous binary classification loss (with constant $K$) replacing the triplet loss, and (b) Embeddings are norm-bounded by $B$. The disciple is a dual encoder with parameters $\theta_D = \{\theta_D^q, \theta_D^l\}$, where $\theta_D^q \in \mathcal{F}$ and $\theta_D^l \in \mathcal{G}$ represent the query and label tower parameters, respectively. The hypothesis classes $\mathcal{F}$ and $\mathcal{G}$ have bounded complexities, characterized by Rademacher constants $R_q$ and $R_l$. The following is our key result:

**Theorem 1.** *Given MOGIC's problem setting, if the disciple model is trained by minimizing the oracle-guided loss $\mathcal{L} = \mathcal{L}_{\text{Alignment}} + K \cdot B \cdot \mathcal{L}_{\text{Matching}}$ on the training set $\mathcal{D} \sim D$ with $N$ samples, then for some $\lambda > 0$ and any $\delta \in [0, 1]$, the following inequality holds true with probability at least $1 - \delta$:*

$$\mathbb{E}_{((X,Z),y)\sim D}\left[\mathcal{L}_{\text{Disciple}}\right] \leq \mathbb{E}_{((X,Z),y)\sim D}\left[\mathcal{L}_{\text{Oracle}}\right]$$
$$+ \frac{4K}{N}\cdot(R_q + R_l) + 2\sqrt{\frac{\log(\frac{1}{\delta})}{N}}$$

*Proof.* The proof is provided in the Appendix E.

The above theorem shows that, post oracle-guided training, the disciple's expected population loss tends to be close to oracle's population loss itself, thus inheriting strong oracle accuracy. This implies good training efficiency for the proposed MOGIC framework.

# 4. Experiments and Results

We now present the experimental setup, and results on training various disciple models using the MOGIC framework.

## 4.1. Datasets and Experimental Setup

The XML Repository (Bhatia et al., 2016) provides various public XC datasets which have been thoroughly studied in the literature (You et al., 2019; Guo et al., 2019; Dahiya et al., 2021; Mittal et al., 2021; Saini et al., 2021; Gupta et al., 2023; Mohan et al., 2024). However, very few of them (Mohan et al., 2024; Chien et al., 2023) offer ground truth metadata. To address this, we attach ground truth metadata from the original dumps to existing XC datasets. Table 9 in Appendix F.1 summarizes the dataset statistics.

The Wikipedia datasets were created from publicly available

Wikipedia dumps [1] dated 2022-05-20. **LF-WikiTitles-500K** and **LF-Wikipedia-500K** (full-text version of the former) are datasets where, given a Wikipedia article/page, the task is to predict the Wikipedia categories that the article should be tagged with. Other Wikipedia article titles connected to the original page via hyperlinks in the article are used as metadata. Similarly, the task in the **LF-WikiSeeAlsoTitles-320K** and **LF-WikiSeeAlso-320K** (full text version of the former) datasets is, given a Wikipedia article/page, to predict the other Wikipedia articles to be recommended in the *"See Also"* section. The Wikipedia categories that these articles are tagged with are used as metadata in this scenario. In a similar vein, the Amazon datasets are created from publicly available Amazon Product review dumps[2]. Given a product title, the task in the **LF-AmazonTitles-131K** dataset and its full-text version, **LF-Amazon-131K** is to predict products that are likely to be bought together. The product category that these products are tagged with serve as the metadata.

**Implementation Details**: We initialize all oracle and disciple encoders with a DistilBERT checkpoint pre-trained on the MS-MARCO dataset (Chen et al., 2024) and fine-tune it. Table 10 in Appendix F.2 summarizes the various hyperparameters used for each dataset. We remark that MOGIC uses golden linkages for the metadata only at training time, whereas at inference time, these metadata linkages are induced (*i.e.,* for a new query, links to metadata are predicted by the disciple). All models were trained using the PyTorch library on a machine with 4 AMD MI200 GPUs.

## 4.2. Results

We now present the experimental results and conduct an ablation study to analyze the contributions of different components of our proposed framework (cf. Appendix G.2 for further details).

**Main results on benchmark datasets**: MOGIC is compared against state-of-the-art XC and dense retrieval approaches in Table 2. MOGIC leads to state-of-the-art accuracy on multiple datasets. These accuracy gains are attributed to gradient regularization from the oracle model.

In particular, MOGIC (OAK) outperforms OAK by 1–2% in P@1 and 2–3% in propensity-scored metrics. In addition to OAK, MOGIC also outperforms graph-based approaches such as GraphFormers (Yang et al., 2021) and GraphSage (Hamilton et al., 2017) by 8%, even those these baseline approaches are given an unfair advantage with access to ground-truth memory items during inference. Note that MOGIC makes no changes to the input, but only the training procedure is improved with the inclusion of a novel

---

[1] `https://dumps.wikimedia.org/enwiki//`

[2] `https://cseweb.ucsd.edu/~jmcauley/datasets.html#amazon_reviews`

Table 2: **Main Results**. Results on public benchmark datasets. MOGIC is up to 2% more accurate as compared to baselines. Results on full-text datasets are in Appendix G.5. For details on evaluation metrics, see Appendix F.4.

| Method | P@1 | P@5 | N@5 | PSP@5 |
|---|---|---|---|---|
| *LF-WikiSeeAlsoTitles-320K* | | | | |
| MOGIC (OAK) | **34.62** | **17.93** | **27.44** | **33.18** |
| OAK | *33.71* | *17.12* | 24.53 | *30.83* |
| DEXA | 32.91 | 16.77 | 24.63 | 29.55 |
| NGAME | 32.64 | 16.6 | 23.44 | 29.87 |
| ANCE | 30.79 | 15.36 | *25.14* | 28.73 |
| DEXML | 29.9 | 14.80 | 22.80 | 25.70 |
| GraphFormers | 21.94 | 11.79 | 24.02 | 22.70 |
| GraphSAGE | 23.13 | 8.26 | 25.12 | 18.73 |
| *LF-WikiTitles-500K* | | | | |
| MOGIC (OAK) | *47.28* | **18.55** | **34.97** | **26.12** |
| OAK | 44.82 | *17.67* | *33.72* | *24.90* |
| DEXA | **47.41** | 17.62 | 33.64 | 24.03 |
| NGAME | 39.04 | 16.08 | 30.75 | 23.03 |
| ANCE | 29.68 | 12.51 | 25.10 | 21.18 |
| GraphFormers | 24.53 | 11.33 | 20.35 | 19.53 |
| GraphSAGE | 21.14 | 11.3 | 22.61 | 11.82 |
| *LF-AmazonTitles-131K* | | | | |
| MOGIC (OAK) | **47.01** | **22.40** | **49.51** | **50.33** |
| OAK | *46.42* | *21.88* | *49.06* | *49.78* |
| DEXA | 46.42 | 21.59 | 49.00 | 49.65 |
| NGAME | 46.01 | 21.47 | 48.67 | 49.43 |

regularization loss. Furthermore, since MOGIC is a regularization framework, it leads to no additional inference cost (cf. Appendix G.3).

Figure 2 presents a quantile-wise comparison of MOGIC and baselines on LF-WikiSeeAlsoTitles-320K. The leftmost bin contains the highest fraction of rare (tail) labels whereas the rightmost bin contains the most popular (head) labels. MOGIC (OAK) gives consistent gains in tail bins and comparable results in head bins (see Appendix F.3 for details).

**Choice of oracle model**: MOGIC's improvement can be attributed to the oracle model it uses to guide the gradients of the disciple. In Table 3, we show results with three different models as the oracle: A finetuned DistilBERT (which is our recommended choice of the Oracle), and two LoRA-finetuned SLMs, Phi-2 and LLaMa-2. A learnable linear projection layer downsizes the SLM embeddings for computing the `Alignment` and `Matching` losses. A disciple regularized with DistilBERT performs on par with the SLM oracles. For all experiments, we use metadata-infused DistilBERT as our oracle due to its faster training and inference times. This choice increases efficiency without sacrificing disciple accuracy. We compare against SLM oracles applied to the task in Appendix G.6

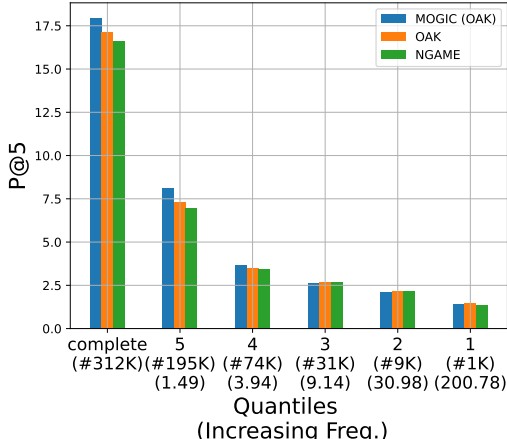

Figure 2: Quantile wise-comparison of MOGIC and baselines for LF-WikiSeeAlsoTitles-320K

Table 3: **Choice of oracle models**. Performance comparison of MOGIC on the LF-WikiSeeAlsoTitles-320K, considering different oracle models. The DistilBERT oracle outperforms the LoRA-finetuned SLM oracles. Nevertheless, via the MOGIC framework, the disciple is capable of leveraging the SLM oracles' signals to improve its task performance.

| Oracle (# params) | Finetuning | P@1 | P@5 | N@5 | PSP@1 | PSP@5 |
|---|---|---|---|---|---|---|
| *MOGIC (OAK)* | | | | | | |
| DistilBERT (65M) | full | *34.62* | *17.93* | *35.70* | **27.44** | **33.18** |
| LLaMA-2 (7B) | LoRA | **34.64** | **17.93** | **35.71** | *27.28* | *33.02* |
| Phi-2 (2.7B) | LoRA | 34.34 | 17.73 | 35.44 | 27.09 | 32.71 |
| *Oracle-only Performance* | | | | | | |
| DistilBERT (65M) | full | **47.63** | **22.75** | **48.37** | **36.71** | **41.45** |
| LLaMA-2 (7B) | LoRA | *34.20* | *16.21* | *33.14* | *30.46* | *31.93* |
| Phi-2 (2.7B) | LoRA | 33.32 | 15.48 | 31.87 | 29.75 | 30.61 |

**Ablations on loss functions**: To understand the importance of each loss term in the second phase of MOGIC training, we ablate by removing each of the three losses, one by one. Lastly, we remove both the oracle-guidance-based loss functions (`Matching` and `Alignment`). The results, presented in Table 4, suggest that all loss components are important and the best accuracy is achieved when all components are used.

**Impact of using ground-truth metadata**: A practical system uses metadata predicted by the disciple, given a query. Hence, all experimental results thus far for MOGIC have been reported considering predicted metadata linkages. In MOGIC we observe a large gap in performance of Oracle and MOGIC (disciple) model, which we attribute to the oracle's access to ground-truth links to the metadata during both training and testing. We therefore hypothesize that in an ideal scenario, using ground-truth metadata at test time could boost overall XC task accuracy. To validate this, we replace predicted metadata in MOGIC with ground-truth metadata during inference and observe that the gap in per-

Table 4: **Ablations on loss functions**. We present ablations on Alignment and Matching losses. The MOGIC (OAK) configuration considers all three loss components: Disciple + Alignment + Matching.

| Loss terms in $\mathcal{L}$ | P@1 | P@5 | N@5 | PSP@1 | PSP@5 |
|---|---|---|---|---|---|
| LF-WikiSeeAlsoTitles-320K | | | | | |
| MOGIC (OAK) | **34.62** | **17.93** | **27.44** | **35.70** | **33.18** |
| Disciple + Alignment | _34.12_ | _17.66_ | 26.72 | 35.16 | _32.57_ |
| Disciple + Matching | 34.11 | 17.63 | _26.83_ | _35.24_ | 32.40 |
| Disciple | 33.71 | 17.12 | 24.53 | 33.83 | 30.83 |
| Alignment + Matching | 32.70 | 16.92 | 26.03 | 33.60 | 31.30 |
| LF-WikiTitles-500K | | | | | |
| MOGIC (OAK) | **47.28** | **18.55** | **34.97** | **27.29** | **26.12** |
| Disciple + Alignment | 45.22 | 17.58 | 33.49 | _27.24_ | _25.10_ |
| Disciple + Matching | _46.03_ | 16.86 | 32.86 | 26.87 | 24.19 |
| Disciple | 44.82 | _17.67_ | _33.72_ | 25.79 | 24.90 |
| Alignment + Matching | 44.93 | 17.40 | 33.18 | 26.87 | 24.73 |

Table 5: **Impact of using ground truth metadata**. We attribute the large gap between the oracle and disciple models' performance to the availability of ground-truth metadata during training and inference. We observe that providing ground-truth metadata at test time to MOGIC (OAK) reduces the gap in performance between the oracle and disciple from 8% to 6%. Additionally MOGIC is robust to noisy (predicted) metadata but the oracle is not. The oracle accuracy drops by 17% in P@1 on LF-WikiSeeAlsoTitles-320K.

| Models | Metadata Source | P@1 | P@5 | N@5 | PSP@1 | PSP@5 |
|---|---|---|---|---|---|---|
| MOGIC (OAK) | Ground-truth | **36.94** | **19.12** | **29.00** | **38.42** | **35.07** |
| | Predicted | _34.62_ | _17.93_ | _27.44_ | _35.70_ | _33.18_ |
| Oracle | Ground-truth | **47.63** | **22.75** | **48.37** | **36.71** | **41.45** |
| | Predicted | _25.09_ | _12.88_ | _19.31_ | _26.05_ | _23.33_ |

formance between the oracle and disciple decreases from 8% to 6% on the LF-WikiSeeAlsoTitles-320K dataset.

The ability to extract useful information from noisy metadata is a desirable quality for a disciple. We can measure this via a model's ability to handle predicted metadata during inference. Table 5 shows that using predicted metadata leads to a decrease of just ∼1–2% for MOGIC (OAK) across metrics, showing that it is fairly robust. However, the oracle accuracy drops by 22% in P@1 when it uses predicted metadata, likely due to the oracle's reliance on ground-truth metadata during training. Although MOGIC receives guidance from a relatively less-robust oracle, the mechanisms in our oracle-guidance-based training enables learning a significantly more robust disciple.

**MOGIC is applicable to any XC disciple**: While we present the main results using OAK as the disciple model, to test the general applicability of the MOGIC framework, we experiment with two other popular XC disciples: NGAME

Table 6: **MOGIC is applicable to any XC disciple**. On LF-WikiSeeAlsoTitles-320K, MOGIC improves accuracy of the base algorithm by 1-2% in P@1. For results on LF-AmazonTitles-131K dataset, refer Appendix G.4.

| Models | P@1 | P@5 | N@5 | PSP@1 | PSP@5 |
|---|---|---|---|---|---|
| MOGIC (OAK) | **34.62** | **17.93** | **27.44** | **35.70** | **33.18** |
| OAK | _33.71_ | _17.12_ | _24.53_ | _33.83_ | _30.83_ |
| MOGIC (NGAME) | **32.37** | **16.38** | **26.87** | **33.16** | **31.08** |
| NGAME | _30.72_ | _15.42_ | _25.18_ | _31.56_ | _28.88_ |
| MOGIC (DEXA) | **32.75** | **16.92** | **26.88** | **34.00** | **31.82** |
| DEXA | _31.57_ | _16.14_ | _25.64_ | _32.71_ | _29.99_ |

Table 7: **Robustness to missing metadata**: Results comparing the performance of MOGIC (OAK) when reducing the size of memory bank on LF-WikiSeeAlsoTitles-320K to simulate missing metadata. **Size (%)** denotes the percentage of the metadata retained. As the memory bank size is decreased by randomly removing items, MOGIC's performance decreases only slightly (both with predicted and with ground truth metadata) but Oracle suffers significantly.

| Size (%) | P@1 | P@5 | N@5 | PSP@1 | PSP@5 |
|---|---|---|---|---|---|
| MOGIC + Predicted metadata | | | | | |
| 100 | **34.62** | **17.93** | **27.44** | **35.70** | **33.18** |
| 80 | _34.54_ | _17.87_ | _27.36_ | _35.60_ | _33.08_ |
| 60 | 34.38 | 17.81 | 27.29 | 35.47 | 32.98 |
| 40 | 34.17 | 17.72 | 27.22 | 35.28 | 32.85 |
| MOGIC + Ground-truth metadata | | | | | |
| 100 | **36.94** | **19.12** | **29.00** | **38.42** | **35.07** |
| 80 | _36.69_ | _19.02_ | _28.82_ | _38.17_ | _34.92_ |
| 60 | 36.47 | 18.86 | 28.70 | 37.85 | 34.66 |
| 40 | 35.91 | 18.59 | 28.45 | 37.24 | 34.26 |
| Oracle | | | | | |
| 100 | **47.63** | **22.75** | **48.37** | **36.71** | **41.45** |
| 80 | _39.22_ | _19.07_ | _30.44_ | _40.15_ | _35.08_ |
| 60 | 35.08 | 17.30 | 27.58 | 36.13 | 32.08 |
| 40 | 30.64 | 15.43 | 24.45 | 31.80 | 28.92 |

and DEXA. Table 6 shows that MOGIC provides 1–2% improvement in precision and NDCG, and 2–3% improvement in PSP over the base XC algorithms.

### 4.3. Robustness Analysis

We now analyze the robustness of the MOGIC framework to both missing metadata and noisy metadata.

**Robustness to missing metadata**: MOGIC uses memory items to improve query representation and regularize XC models. To simulate missing metadata, we decrease the size of the memory bank by randomly removing items (*i.e.,* by randomly subsampling the set of retrieved memory items). In Table 7, we present the performance of MOGIC (OAK) when the size of the memory bank is reduced. We observe

Table 8: **Robustness to missing metadata**: MOGIC is more robust to noisy metadata that the oracle it is guided by. Introducing noise in the fused metadata at inference time can lead to up to 20% reduction in accuracy of the oracle, since early-fusion models rely on high-quality metadata at the input unlike the late-fusion-based MOGIC model. The performance of MOGIC (OAK) decreases only slightly, indicating that models trained with MOGIC are robust to noise in the metadata.

| Noise% | P@1 | P@5 | N@5 | PSP@1 | PSP@5 |
|---|---|---|---|---|---|
| | MOGIC (OAK) | | | | |
| 0 | **36.94** | **19.12** | **29.00** | **38.42** | **35.07** |
| 20 | *36.26* | *18.80* | *28.66* | *37.69* | *34.61* |
| 40 | 35.62 | 18.44 | 28.36 | 36.90 | 34.08 |
| 60 | 34.92 | 18.12 | 27.94 | 36.19 | 33.59 |
| | Oracle | | | | |
| 0 | **47.63** | **22.75** | **48.37** | **36.71** | **41.45** |
| 20 | *34.80* | *16.83* | *26.67* | *35.64* | *30.73* |
| 40 | 26.75 | 13.1 | 20.45 | 27.56 | 23.87 |
| 60 | 18.65 | 9.31 | 14.29 | 19.44 | 17.02 |

that, as the size of memory bank is decreased, MOGIC's performance decreases only marginally (both with predicted as well as with ground-truth metadata) but the oracle performance worsens significantly. We observe that the drop in accuracy is larger for the oracle model in comparison to MOGIC, which we attribute to MOGIC's robustness to noise in the memory items.

**Robustness to noisy metadata**: To further understand the relationship between the oracle's and MOGIC's performance under the action of noise, we consider the following setting. First, while predicting, we take the ground-truth metadata from both the oracle and the MOGIC models. Subsequently, for every query, we inject varying levels of noise (from 0% to 60%) to the ground-truth metadata, and measure the degree of robustness of both the models to such noise. Noise is added by randomly replacing a certain percentage of ground-truth metadata items with irrelevant ones. Table 8 shows that, as we increase noise, the XC task performance decreases for both the models, signifying the importance of clean metadata. However, MOGIC's downstream performance decreases only slightly, while the oracle's performance decreases significantly, demonstrating the robustness of MOGIC. We also observe that MOGIC (OAK) is robust to noise in the metadata even during training, and the drop in its performance is negligible. For additional details, refer to Appendix G.7.

## 5. Conclusion

We introduce MOGIC, a novel framework for enriching query representations using relevant metadata without incur-

ring high inference latency. This is achieved via a two-phase training. The first phase trains an oracle using textual metadata infusion both on the query as well as the label side. The second phase involves guiding the training of a disciple model using embeddings from the oracle classifier.

A key reason for the effectiveness of our method is that MOGIC, trained with the proposed regularization strategy, effectively retains the original intention of the query. It leverages the best of both worlds: maintaining low inference latency and gracefully handling noisy or missing metadata via late-fusion, while simultaneously learning to mimic the superior embeddings of the oracle through our regularization loss. This combined approach mitigates the respective drawbacks of both late- and early-fusion strategies, leading to consistently superior performance.

Through extensive experimentation on six popular benchmark XC datasets, we demonstrated that the MOGIC significantly outperforms state-of-the-art XC models, achieving improvements in terms of precision, NDCG, and propensity-scored precision. Moreover, MOGIC exhibits robustness to missing and noisy metadata, making it a valuable tool for real-world applications.

In conclusion, MOGIC represents a significant advancement in the field of extreme classification, offering a practical and effective solution for incorporating metadata to enhance performance. Our work highlights the potential of oracle-guided training in improving the robustness and accuracy of memory-based models in challenging classification tasks.

## Acknowledgments

We gratefully acknowledge the financial support provided by the Yardi School of Artificial Intelligence at the Indian Institute of Technology Delhi, and the Prime Minister's Research Fellowship. We also thank HPC, IIT Delhi and Microsoft Research India for providing computational resources.

## Impact Statement

Our usage of data and terms of providing service to people around the world has been approved by our legal and ethical boards. In terms of social relevance, our research is helping millions of people find the goods and services that they are looking for online with increased efficiency and a significantly improved user experience, both via sponsored search ads as well as via e-commerce search. This facilitates purchase and delivery without any physical contact which is important given today's social constraints. Furthermore, our research is increasing the revenue of many small and medium businesses including mom and pop stores while also helping them grow their market and reduce the cost of reaching new customers.

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

# Appendix

## Table of Contents

# A. Preliminaries

- **Ground-truth metadata:** Besides the query text, often, a variety of auxiliary information is available in many domains, e.g., frequently clicked webpages for search queries in sponsored search, previously searched queries for web search query auto-completion, etc. Auxiliary information available from disparate but related tasks often have relevant diverse information that the input query does not, which can be leveraged to provide better predictions. We call such auxiliary information as ground-truth metadata. For example, on sponsored search ads task that involves query-to-ad-keyword prediction, the query-side metadata is obtained by mining the organic search webpage titles clicked in response to the query on the search engine, while on the Wikipedia categories prediction task, other Wikipedia article titles connected to the original page via hyperlinks could serve as the metadata.

- **Early-fusion**: When the query and metadata tokens are concatenated in the original text form itself (initial stage of processing) rather than in embedding form, we call it early-fusion. This approach contrasts with late-fusion, where these are combined at later stages.

- **Late-stage fusion**: When the query and metadata embeddings are combined after their tokens have been processed through multiple Transformer encoder layers, we call this combination as late-stage fusion.

- **Oracle**: An oracle is a model with access to metadata information during both training and inference. These models are characterized by their superior metric values, which is their sole purpose, without any consideration for model size or computational cost (measured in FLOPs). Due to their impractical computational demands, oracles cannot be deployed in real-world applications.

- **Disciple**: A disciple is a frugal model that attempts to mimic the performance of an oracle model. While not as superior as an oracle, a disciple excels in terms of model size and computational cost, making it suitable for deployment in real-world applications.

- **Metadata-infused oracle**: This is an oracle model which is given access to ground truth metadata. Metadata-infused Oracle is the crux of our MOGIC framework. In the two stage method, Metadata-infused oracle training forms the first stage. In this stage, an oracle model is trained using both query-side and label-side ground-truth metadata. This metadata is in textual form and is used to enhance the training process.

- **Memory-based models versus Memory-free models**: Memory-based XC models are models that have access to memory (metadata). Very few XC methods are memory-based. On the other hand, most of the XC methods do not leverage metadata at all and are therefore called memory-free methods.

- **Query-side metadata**: Additional auxiliary information related to the input query is called query-side metadata. For example, in Table 1, for the query "Grass court", query-side metadata can be "Tennis terminology", "Sports rules and regulations", "Tennis court surfaces".

- **Label-side metadata**: Additional auxiliary information related to a label is called label-side metadata. For example, for the label "Clay court", label-side metadata is "Tennis terminology", "Sports rules and regulations", "Clay", "Tennis court surfaces".

- **Missing labels**: XC tasks involve a vast number of labels, making it impractical for annotators to mark all potential labels. This inherent limitation often results in missing labels, which are labels that should have been included in the ground truth but were inadvertently omitted.

- **Memory bank**: The memory bank is a collection of vector embeddings of all metadata associated with both the queries and labels, parameterized by $\theta_M$. Formally, memory bank is represented by $\mathcal{M} \in \mathbb{R}^{M \times D}$, each memory item $j$ is mapped to a row in the matrix which we call its memory item representation $\mathbf{m}_j \in \mathbb{R}^D$. Here $\mathcal{M}(\cdot|\theta_M)$ returns relevant memory items for query $X_q$ and label $Z_l$, i.e., $\mathbf{m}_{q/l} \in \mathcal{M}(X_q/Z_l|\theta_M)$.

- **Rademacher complexity**: Rademacher complexity constants $R_q, R_l$ are estimated as the average empirical loss of minimizing the hypothesis class on a data sample with randomly annotated labels i.e. labels are generated by a purely random Bernoulli distribution with probabilities 0.5 to be either positive or negative. In intuitive terms, the smaller the values of $R_q, R_l$, the less prone are the query tower and label tower to overfit the finite training data, and consequently the accuracy on the test set is expected to be better.

## B. Visualization of MOGIC (OAK)

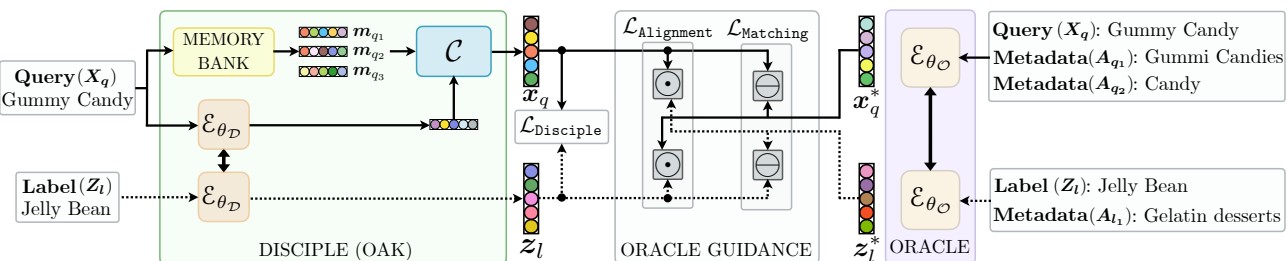

Figure 3: MOGIC (OAK) training framework, depicted in Figure 1, is presented to facilitate a detailed explanation.

In this figure, we explain the detailed architecture of the proposed MOGIC framework. The input query is "Gummy Candy", and the expected label is "Jelly Bean". The overall framework comprises of 4 different parts: (1) disciple, (2) task specific loss functions, (3) guidance from the oracle to the disciple, and (4) the oracle itself. The framework comprises of 2 stages: oracle training and disciple training. The oracle training leverages both the query-side as well as the label-side ground truth metadata. As shown in the Figure 3, the query-side ground truth metadata includes 2 memory items: "Gummi Candies" and "Candy". The label side ground truth metadata includes just one memory item in this case: "Gelatin Desserts". This rich metadata is concatenated with the query and the label in the text form to train the oracle. In stage 2, we train the disciple based on guidance from the oracle. This stage 2 is illustrated in detail in the Figure 3. The disciple denoted by the green box, consists of an encoder, memory bank and a combiner $\mathcal{C}$. At train time, the disciple first encodes both the query as well as the label using the same encoder. Since the disciple has to be low latency, it must involve late fusion. For late fusion, we need embedding representations of both the query as well as the metadata items. To obtain embedding of the metadata items, the query is also sent to the memory bank. The combiner actually performs the late fusion using cross attention and outputs an enhanced query representation. Of course while training the disciple, we need to ensure that the task specific loss is minimized. This loss tries to maximize the similarity between the embedding of the label and the enhanced query representation, and is illustrated in the task part of the Figure 3. The guidance part of the Figure 3 shows how the guidance is passed on by the oracle to the disciple using 4 different loss functions. These loss functions try to maximize the similarity between (a) oracle's query representation and the disciple's query representation (b) oracle's label representation and the disciple's label representation (c) oracle's query representation and the disciple's label representation, and (d) oracle's label representation and the disciple's query representation.

## C. End-to-end Walkthrough of the MOGIC Framework

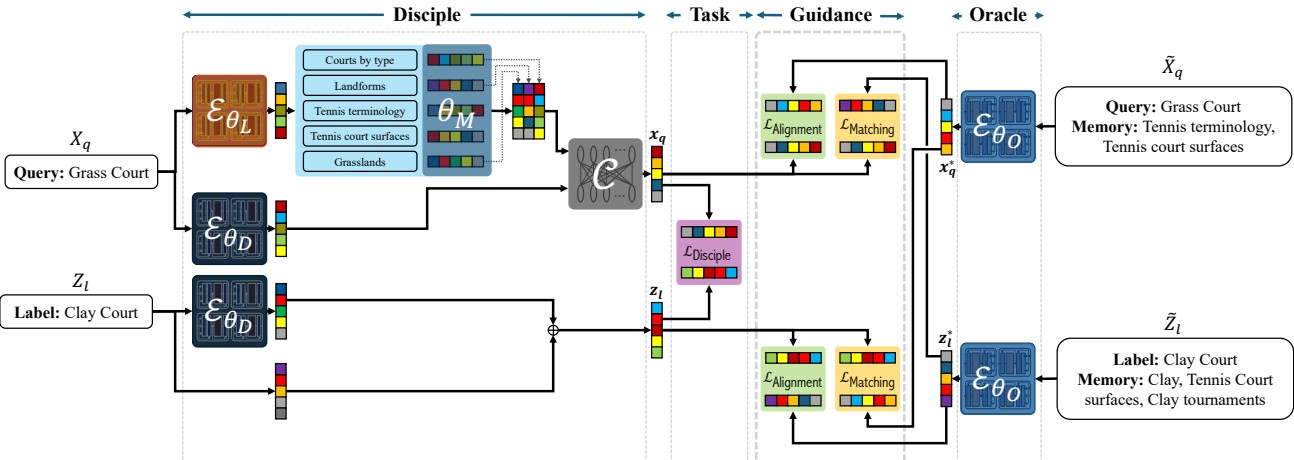

Figure 4: This is a detailed version of Figure 1. $\theta_L$ represents the parameters of the classifier that predicts metadata for a given query. $\theta_M$ denotes the memory bank containing memory items corresponding to query and label metadata. $\theta_D$ represents the parameters of the base encoder in disciple. $\mathcal{C}$ signifies the combiner module. Collectively, $\theta_L$, $\theta_M$, $\theta_D$, and $\mathcal{C}$ constitute the parameters of the disciple, denoted as $\theta_{\mathcal{D}}$.

This section provides a more detailed walkthrough of the MOGIC framework as depicted in Figure 1. Figure 4 shows a more detailed version of the architecture and it will help us to walk through an example on the training and the inference of the MOGIC framework. Let us consider the query "Grass Court" mentioned in Table 1. The framework has three major blocks – query processing block, label processing block and oracle representation block. The query processing block and label processing block are part of the disciple.

**Query processing block:** involves the following steps

- **Metadata retrieval:** The query "Grass Court" is sent to the memory bank to retrieve relevant metadata, including "Courts by type", "Landforms", and "Grasslands". The memory bank contains vector representations for each of these metadata. These vector representations are then sent for further processing.
- **Query encoding:** The query "Grass court" is passed through the disciple encoder to obtain its vector representation.
- **Query enrichment:** The query representation is fused with the metadata representation using a cross-attention layer to create an enriched query representation.

**Label processing block:** similar to the above query processing

- **Label encoding:** The label "Clay Court" is passed through the base encoder to obtain its vector representation. This encoder is shared between query and the label.
- **Label enrichment:** The label vector representation is further enriched by combining it with a separate "Clay Court" free parameter.

**Oracle representation block:** The query "Grass Court" is concatenated with its associated ground truth metadata "Tennis terminology", "Sports rules and regulations", "Tennis court surfaces" to form a super query "Clay Court Tennis terminology Sports rules and regulations Tennis court surfaces". This super query is passed through the oracle encoder to obtain its vector representation. Similarly "Clay court" is concatenated with "Clay", "Tennis court surfaces" and "Clay tournaments" and passed to the shared oracle encoder to obtain its vector representation.

The above blocks are then used for training and inference,

**Inference:** During inference for the query "Grass Court" we compute its query representation using the query processing block and then calculate its similarity with all label representations, present in the dataset, computed from the label processing

block, including "Alabama", "Clay Court", "Carpet Court", "Hardcourt" and "Henry Moore", to determine their relevance to the query using cosine similarity distance.

**Training:** Unlike inference, training uses all the blocks and involves the following steps:

- **Vector representations:** We compute query and label representations for both the query "Grass Court" and the label "Clay Court" using the query processing block and the label processing block.

- **Triplet loss:** We then apply triplet loss to the query and label representations, as is common in retrieval methods.

- **Oracle regularization:** To further regularize the disciple model, we introduce additional triplet loss terms (a) between the query "Grass Court" oracle representation and the "Clay Court" label representation. (b) between the label "Clay Court" oracle representation and the "Grass Court" query representation. (c) mean squared error (MSE) loss to minimize the distance between the "Grass Court" oracle representation and its query representation and the "Clay Court" oracle representation and its label representation.

**What makes MOGIC (OAK) perform better than OAK?**

We observe that for the first example "Grass Court", "Courts by type" is a good memory item, but some predicted memory items ("Landforms" and "Grasslands") are misleading. This is the same metadata which is used both by OAK as well as MOGIC (OAK). Unfortunately, the misleading metadata causes OAK to produce bad predictions about geographical places like "Texas, List of Nevada state prisons, Ronald Reagan Boyhood Home, West End (Richmond, Virginia)". The oracle-guidance fortunately helped MOGIC (OAK) to avoid paying attention to the misleading metadata and therefore MOGIC (OAK) ends up predicting accurate labels like "Clay court, Carpet court, Hardcourt". Even the label "U.S. Men's Clay Court Championships" is somewhat relevant. With MOGIC regularized training, MOGIC (OAK) was able to retain the original intent of the query and predicted various type tennis courts.

### C.1. Relation between MOGIC and Knowledge Distillation

The disciple training in MOGIC is a novel variant of knowledge distillation (KD). Compared to standard KD, it differs in multiple ways. First, the metadata is provided as early concatenation in textual form for the oracle, but is used to train free parameters in the disciple's memory via a novel regularization framework. This can be viewed as knowledge distillation from an early-concatenation model to a two-tower model. While all disciple models benefit from this framework, the additional parameters present in the memory-based disciples such as MOGIC (OAK) demonstrate the most gains. Furthermore, the oracle has access to ground-truth metadata, which is privileged information that is not available to the disciple model (unlike standard KD). In this MOGIC framework, the oracle can either be larger, or of the same size as the disciple. Table 3 presents comparisons between larger (LLM-based, 2.7B/7B sized) oracle and a disciple-sized (65M sized), DistilBERT oracle. For more details, please refer to Section 4.2.

## D. Prompt for SLMs

```
Given the title of a wikipedia article and the corresponding categories of that article on wikipedia, your task is
to predict the titles of all articles which are likely to be listed in the see also section of the mentioned
article. Output the coma separated list of titles of the articles in the see also section of the given article.

\#\#\# Input : \newline
\#\#\# Title : agricultural science \newline
\#\#\# Categories : agriculture, agronomy \newline

\#\#\#\# Task Output \newline
\#\#\#\# Predicted titles \newline
agricultural sciences basic topics, agriculture ministry, agroecology, american society of agronomy, genomics of
domestication, history of agricultural science, institute of food and agricultural sciences, international
assessment of agricultural science and technology for development, national ffa organization, agricultural science.
```

Listing 1: This prompt aims to fine-tune small language models (LLaMA-2 and Phi-2) using LoRA to predict the titles of articles likely to appear in the 'See Also' section of a Wikipedia article given its title and associated categories.

# E. Theoretical Proofs

## E.1. Notations

Let $\mathfrak{D} = \left\{ \{(X_i, \mathbf{y}_i)\}_{i=1}^N, \{Z_l\}_{l=1}^L \right\}$ be the training dataset, where $X_i, Z_l$ are the raw text features of $i^{\text{th}}$ query and $l^{\text{th}}$ label respectively, and $\mathbf{y}_i \in \{0,1\}^L$ is the binary label vector for the $i^{\text{th}}$ query.

Let the oracle be a dual encoder model denoted by parameters $\theta_O = \{\theta_O^q, \theta_O^l\}$. Similarly, let the disciple also be a dual encoder model denoted by parameters $\theta_D = \{\theta_D^q, \theta_D^l\}$. Given the raw text query and label samples $X$ and $Z$, let the frozen oracle embeddings be denoted as $\mathbf{x}^* = \mathcal{E}_{\theta_O}(X), \mathbf{z}^* = \mathcal{E}_{\theta_O}(Z)$, and the trainable disciple embeddings be denoted by $\mathbf{x} = \mathcal{E}_{\theta_D}(X), \mathbf{z} = \mathcal{E}_{\theta_D}(Z)$.

Let $\mathbf{Z} = \{\mathbf{z}_1, \ldots, \mathbf{z}_L\}$ be the matrix of all label embeddings stacked together. Consider the loss $\mathcal{L}(\{(\mathbf{x}_i, \mathbf{y}_i)\}_{i=1}^N, \{\mathbf{z}_l\}_{l=1}^L) = \frac{1}{N} \sum_{i=1}^N \ell(\mathbf{Z}^\top \mathbf{x}_i, \mathbf{y}_i)$ to be a generic loss function that is separable over query samples. Note that the triplet loss used by MOGIC falls in this family of loss functions, and therefore the analysis presented below holds true for it.

Let the query and label towers of the disciple model $\theta_D = \{\theta_D^q, \theta_D^l\}$ belong to hypothesis classes $\mathcal{F}, \mathcal{G}$ whose complexities be bounded by Rademacher constants $R_q, R_l$ respectively.

The Rademacher complexity constants $R_q$ and $R_l$ are scalar values that quantify the complexity or capacity of the hypothesis classes corresponding to the query tower and the label tower, respectively. We use the standard definitions of Rademacher constants from the Statistical Learning Theory for Binary Classification problems [3] [4].

Under the above setting, the `Alignment` and `Matching` losses can be expressed as:

$$\mathcal{L}_{\texttt{Alignment}} = \frac{1}{2} \Big( \mathcal{L}(\{(\mathbf{x}_i, \mathbf{y}_i)\}_{i=1}^N, \{\mathbf{z}_l^*\}_{l=1}^L) + \mathcal{L}(\{(\mathbf{x}_i^*, \mathbf{y}_i)\}_{i=1}^N, \{\mathbf{z}_l\}_{l=1}^L) \Big) \tag{4}$$

$$\mathcal{L}_{\texttt{Matching}} = \frac{1}{N} \sum_{i=1}^N \|\mathbf{x}_i - \mathbf{x}_i^*\|_2 + \frac{1}{L} \sum_{l=1}^L \|\mathbf{z}_l - \mathbf{z}_l^*\|_2 \tag{5}$$

## E.2. Derivations

The below lemma shows that the triplet loss used in MOGIC is upper-bounded by a linear combination of the `Alignment` and `Matching` loss, assuming that the individual pairwise loss terms comprising the triplet loss are Lipschitz continuous by themselves.

**Lemma 2.** *Let $P$ be the number of positive labels for a given query and, $\ell(\mathbf{s}_i, \mathbf{y}_i) = \frac{1}{P \cdot (L-P)} \sum_{p,q \in \{1,\ldots,L\}} \mathbb{1}(y_{ip} = +1)\mathbb{1}(y_{in} = -1)g(s_{in} - s_{ip})$ be the triplet loss for a query $i$ which is decomposable over all relevant-irrelevant label pairs. If $g$ is Lipschitz-continuous with constant $K$, then the following inequalities hold true:*

$$\mathcal{L}(\{(\mathbf{x}_i, \mathbf{y}_i)\}_{i=1}^N, \{\mathbf{z}_l\}_{l=1}^L) \leq \mathcal{L}(\{(\mathbf{x}_i, \mathbf{y}_i)\}_{i=1}^N, \{\mathbf{z}_l^*\}_{l=1}^L) + \frac{2 \cdot K \cdot B}{L} \sum_{l=1}^L \|\mathbf{z}_l - \mathbf{z}_l^*\|_2 \tag{6}$$

$$\mathcal{L}(\{(\mathbf{x}_i, \mathbf{y}_i)\}_{i=1}^N, \{\mathbf{z}_l\}_{l=1}^L) \leq \mathcal{L}(\{(\mathbf{x}_i^*, \mathbf{y}_i)\}_{i=1}^N, \{\mathbf{z}_l\}_{l=1}^L) + \frac{2 \cdot K \cdot B}{N} \sum_{i=1}^N \|\mathbf{x}_i - \mathbf{x}_i^*\|_2 \tag{7}$$

---

[3] https://web.eecs.umich.edu/~cscott/past_courses/eecs598w14/notes/10_rademacher.pdf
[4] https://www.cs.cmu.edu/~ninamf/ML11/lect1117.pdf

*Proof.* Let $\mathbf{s}_i, \mathbf{s}_i^{'}$ be two score vectors. Then,

$$|\ell(\mathbf{s}_i, \mathbf{y}_i) - \ell(\mathbf{s}_i^{'}, \mathbf{y}_i)| \tag{8}$$

$$= |\frac{1}{P \cdot (L - P)} \sum_{p,q \in \{1,...,L\}} \mathbb{1}(y_{ip} = +1)\mathbb{1}(y_{in} = -1)(g(s_{in} - s_{ip}) - g(s_{in}^{'} - s_{ip}^{'}))| \tag{9}$$

$$\leq \frac{1}{P.(L - P)} \sum_{p,q \in \{1,...,L\}} \mathbb{1}(y_{ip} = +1)\mathbb{1}(y_{in} = -1)|g(s_{in} - s_{ip}) - g(s_{in}^{'} - s_{ip}^{'})| \tag{10}$$

$$\leq \frac{K}{P \cdot (L - P)} \sum_{p,q \in \{1,\cdots,L\}} \mathbb{1}(y_{ip} = +1)\mathbb{1}(y_{in} = -1)(|s_{in} - s_{in}^{'}| + |s_{ip} - s_{ip}^{'}|) \tag{11}$$

If $\mathbf{s}_i = \mathbf{Z}^\top \mathbf{x}_i$ and $\mathbf{s}_i^{'} = \mathbf{Z}^{*\top} \mathbf{x}_i$, then:

$$\frac{1}{N} \sum_{i=1}^{N} |\ell(\mathbf{s}_i, \mathbf{y}_i) - \ell(\mathbf{s}_i^{'}, \mathbf{y}_i)| \tag{12}$$

$$\leq \frac{K \cdot B}{N \cdot P \cdot (L - P)} \sum_{p,q \in \{1,...,L\}} \mathbb{1}(y_{ip} = +1)\mathbb{1}(y_{in} = -1)(\|\mathbf{z}_n - \mathbf{z}_n^*\|_2 + \|\mathbf{z}_p - \mathbf{z}_p^*\|_2) \tag{13}$$

$$\leq \frac{K \cdot B}{N \cdot P \cdot (L - P)} \cdot \frac{2NP(L - P)}{L} \sum_{l=1}^{L} \|\mathbf{z}_l - \mathbf{z}_l^*\|_2 \tag{14}$$

$$= \frac{2 \cdot K \cdot B}{L} \sum_{l=1}^{L} \|\mathbf{z}_l - \mathbf{z}_l^*\|_2 \tag{15}$$

As a result, the following inequality holds true:

$$\mathcal{L}(\{(\mathbf{x}_i, \mathbf{y}_i)\}_{i=1}^{N}, \{\mathbf{z}_l\}_{l=1}^{L}) \leq \mathcal{L}(\{(\mathbf{x}_i, \mathbf{y}_i)\}_{i=1}^{N}, \{\mathbf{z}_l^*\}_{l=1}^{L}) + \frac{2 \cdot K \cdot B}{L} \sum_{l=1}^{L} \|\mathbf{z}_l - \mathbf{z}_l^*\|_2 \tag{16}$$

Similarly, if $\mathbf{s}_i = \mathbf{Z}^\top \mathbf{x}_i$ and $\mathbf{s}_i^{'} = \mathbf{Z}^\top \mathbf{x}_i^*$, then:

$$\frac{1}{N} \sum_{i=1}^{N} |\ell(\mathbf{s}_i, \mathbf{y}_i) - \ell(\mathbf{s}_i^{'}, \mathbf{y}_i)| \tag{17}$$

$$\leq \frac{2 \cdot K \cdot B}{N \cdot P \cdot (L - P)} \sum_{p,q \in \{1,...,L\}} \mathbb{1}(y_{ip} = +1)\mathbb{1}(y_{in} = -1)\|\mathbf{x}_i - \mathbf{x}_i^*\|_2 \tag{18}$$

$$= \frac{2 \cdot K \cdot B}{N} \sum_{i=1}^{N} \|\mathbf{x}_i - \mathbf{x}_i^*\|_2 \tag{19}$$

As a result, the following inequality holds true as well:

$$\mathcal{L}(\{(\mathbf{x}_i, \mathbf{y}_i)\}_{i=1}^{N}, \{\mathbf{z}_l\}_{l=1}^{L}) \leq \mathcal{L}(\{(\mathbf{x}_i^*, \mathbf{y}_i)\}_{i=1}^{N}, \{\mathbf{z}_l\}_{l=1}^{L}) + \frac{2 \cdot K \cdot B}{N} \sum_{i=1}^{N} \|\mathbf{x}_i - \mathbf{x}_i^*\|_2 \tag{20}$$

$$\square$$

**Lemma 3.** *Assume a realizable setting where, for some $\theta_D = \theta_D^*$, $\mathbf{x} = \mathbf{x}^*, \mathbf{z} = \mathbf{z}^*$ holds for all $\mathbf{x}, \mathbf{z}$. Now, let $\theta_D = \bar{\theta}_D$ be another value of $\theta_D$ which minimizes the oracle-guided population loss $\mathcal{L}_{\texttt{Alignment}} + \lambda \mathcal{L}_{\texttt{Matching}}$. Its corresponding embeddings are denoted by $\bar{\mathbf{x}}, \bar{\mathbf{z}}$. Further, assume that all the embeddings are bounded by $\|\bar{\mathbf{x}}\|_2, \|\bar{\mathbf{z}}\|_2, \|\mathbf{x}^*\|_2, \|\mathbf{z}^*\|_2 \leq B$.*

*Then, for $\lambda = K \cdot B$, the following inequalities hold:*

$$\mathcal{L}(\{(\bar{\mathbf{x}}_i, \mathbf{y}_i)\}_{i=1}^{N}, \{\bar{\mathbf{z}}_l\}_{l=1}^{L}) \leq \mathcal{L}_{\texttt{Alignment}} + \lambda \cdot \mathcal{L}_{\texttt{Matching}} \leq \mathcal{L}(\{(\mathbf{x}_i^*, \mathbf{y}_i)\}_{i=1}^{N}, \{\mathbf{z}_l^*\}_{l=1}^{L}) \tag{21}$$

*Proof.* By averaging the two inequalities in Lemma 2, we get the following result:

$$\mathcal{L}(\{(\bar{\mathbf{x}}_i, \mathbf{y}_i)\}_{i=1}^N, \{\bar{\mathbf{z}}_l\}_{l=1}^L) \tag{22}$$

$$\leq \frac{1}{2}(\mathcal{L}(\{(\bar{\mathbf{x}}_i, \mathbf{y}_i)\}_{i=1}^N, \{\mathbf{z}_l^*\}_{l=1}^L) + \mathcal{L}(\{(\mathbf{x}_i^*, \mathbf{y}_i)\}_{i=1}^N, \{\bar{\mathbf{z}}_l\}_{l=1}^L))$$

$$+ K \cdot B(\frac{1}{L}\sum_{l=1}^L \|\bar{\mathbf{z}}_l - \mathbf{z}_l^*\|_2 + \frac{1}{N}\sum_{i=1}^N \|\bar{\mathbf{x}}_i - \mathbf{x}_i^*\|_2) \tag{23}$$

$$= \mathcal{L}_{\text{Alignment}} + K \cdot B \cdot \mathcal{L}_{\text{Matching}} \tag{24}$$

Next, note that $\bar{\mathbf{x}}, \bar{\mathbf{z}}$ are the values of $\mathbf{x}, \mathbf{z}$ which minimize the oracle-guided loss. Due to this and the realizable setting assumption:

$$\mathcal{L}_{\text{Alignment}} + K \cdot B \cdot \mathcal{L}_{\text{Matching}} \tag{25}$$

$$= \frac{1}{2}(\mathcal{L}(\{(\bar{\mathbf{x}}_i, \mathbf{y}_i)\}_{i=1}^N, \{\mathbf{z}_l^*\}_{l=1}^L) + \mathcal{L}(\{(\mathbf{x}_i^*, \mathbf{y}_i)\}_{i=1}^N, \{\bar{\mathbf{z}}_l\}_{l=1}^L))$$

$$+ K \cdot B(\frac{1}{L}\sum_{l=1}^L \|\bar{\mathbf{z}}_l - \mathbf{z}_l^*\|_2 + \frac{1}{N}\sum_{i=1}^N \|\bar{\mathbf{x}}_i - \mathbf{x}_i^*\|_2) \tag{26}$$

$$\leq \frac{1}{2}(\mathcal{L}(\{(\mathbf{x}_i^*, \mathbf{y}_i)\}_{i=1}^N, \{\mathbf{z}_l^*\}_{l=1}^L) + \mathcal{L}(\{(\mathbf{x}_i^*, \mathbf{y}_i)\}_{i=1}^N, \{\mathbf{z}_l^*\}_{l=1}^L))$$

$$+ K \cdot B(\frac{1}{L}\sum_{l=1}^L \|\mathbf{z}_l^* - \mathbf{z}_l^*\|_2 + \frac{1}{N}\sum_{i=1}^N \|\mathbf{x}_i^* - \mathbf{x}_i^*\|_2) \tag{27}$$

$$= \mathcal{L}(\{(\mathbf{x}_i^*, \mathbf{y}_i)\}_{i=1}^N, \{\mathbf{z}_l^*\}_{l=1}^L) \tag{28}$$

The above proves the two inequalities. □

However, optimizing the population-level oracle-guided loss is not feasible as we are often restricted to a finite training sample size. Now, empirical loss optimization on the finite training set introduces some error. The following lemma bounds this error:

**Lemma 4.** *Let the disciple model be trained by minimizing the oracle-guided loss on the training set* $\mathfrak{D} = \{\{(X_i, \mathbf{y}_i)\}_{i=1}^N, \{Z_l\}_{l=1}^L\}$. *Let the empirical training risk attained by this minimization be* $\hat{\mathcal{L}}$, *then the following inequality holds:*

$$|\min_{\theta_D} \mathbb{E}(\mathcal{L} - \hat{\mathcal{L}})| \leq \frac{2K}{N} \cdot (R_q + R_l) + \sqrt{\frac{\log(\frac{1}{\delta})}{N}} \tag{29}$$

*where* $\mathcal{L}$ *is the population-level oracle-guided training loss.*

*Proof.* Proof uses the standard ideas of ghost sampling and Rademacher complexity bounding, along with some well-known properties of Rademacher complexity. Note here that $\min_{\theta_D} \mathbb{E}\mathcal{L} = \mathcal{L}_{\text{Alignment}} + \lambda \cdot \mathcal{L}_{\text{Matching}}$ with $\bar{\mathbf{x}}, \bar{\mathbf{z}}$ embeddings, thus connecting to Lemma 3. □

**Theorem 5.** *Given the problem setting described above, if the disciple model is trained by minimizing the oracle-guided loss* $\mathfrak{L} = L_{\text{Alignment}} + \lambda \cdot \mathcal{L}_{\text{Matching}}$ *on the training set* $\mathfrak{D} = \{\{(X_i, \mathbf{y}_i)\}_{i=1}^N, \{Z_l\}_{l=1}^L\}$, *then for some* $\lambda > 0$ *and any* $\delta \in [0, 1]$, *the following inequality holds true with probability at least* $1 - \delta$:

$$\mathbb{E}_{(\mathbf{x},\mathbf{y})}\ell(\mathbf{Z}^\top \mathbf{x}, \mathbf{y}) \leq \mathbb{E}_{(\mathbf{x},\mathbf{y})}\ell(\mathbf{Z}^{*\top}\mathbf{x}^*, \mathbf{y}) + \frac{4K}{N} \cdot (R_q + R_l) + 2\sqrt{\frac{\log(\frac{1}{\delta})}{N}} \tag{30}$$

*Proof.* Proof involves a simple algebraic combination of the results in Lemmas 3 and 4. □

# F. Experimental Setup

In this appendix, we discuss details regarding the dataset statistics, training hyper-parameters, and the evaluate metrics used.

## F.1. Dataset Statistics

Table 9: A summary of the dataset statistics in terms of the queries (Q), labels (L) and memory items (M). The Avg. queries per label is computed as the average value of the number of positive labels associated with each query in the dataset. Similarly, to find the Avg. labels per query, given a label, we identify the number of queries for which that label is a positive, and compute the average of those numbers. For LF-WikiSeeAlsoTitles-320K and LF-WikiSeeAlso-320K tasks, the Wikipedia categories that these articles are tagged with are used as metadata. For LF-WikiTitles-500K and LF-Wikipedia-500K tasks, the Wikipedia article titles connected to each original page via hyperlinks in the article are used as metadata. For LF-AmazonTitles-131K and LF-Amazon-131K tasks, categories associated with the product are metadata.

| Dataset | # Train Queries (Q) | # Labels (L) | # Test Queries | Avg. Queries/label | Avg. Labels/query | Metadata Types | # Metadata (M) | Avg. Metadata/Query |
|---|---|---|---|---|---|---|---|---|
| LF-WikiSeeAlsoTitles-320K | 693K | 312K | 177K | 2.11 | 4.67 | category | 656K | 4.89 |
| LF-WikiSeeAlso-320K | 693K | 312K | 177K | 2.11 | 4.67 | category | 656K | 4.89 |
| LF-WikiTitles-500K | 1.8M | 501K | 783K | 4.74 | 17.15 | hyper-link | 2.1M | 15.95 |
| LF-Wikipedia-500K | 1.8M | 501K | 783K | 4.74 | 17.15 | hyper-link | 2.1M | 15.95 |
| LF-AmazonTitles-131K | 294K | 131K | 134K | 5.15 | 2.29 | category | 210K | 4.64 |
| LF-Amazon-131K | 294K | 131K | 134K | 5.15 | 2.29 | category | 210K | 4.64 |

## F.2. Hyper-parameters and Training Details

Table 10 shows hyper-parameters used in MOGIC to regularize XC models. SLMs were obtained from the HuggingFace model repository. Phi-2 (2.7B parameters) was retrieved from `https://huggingface.co/microsoft/phi-2`, and LLaMA-2 (7B parameters) was retrieved from `https://huggingface.co/meta-llama/Llama-2-7b`.

Table 10: Hyper-parameter values for MOGIC on all datasets to enable reproducibility. MOGIC code will be released publicly. Most hyperparameters were set to their default values across all datasets. **LR** is learning rate. Margin $\gamma = 0.3$ was used for contrastive loss. A cell containing the symbol $\uparrow$ indicates that that cell contains the same hyperparameter value present in the cell directly above it.

| **Dataset** | **Batch Size** $S$ | **Encoder** epochs | **Encoder LR** $lr$ | **Bert seq. length** $L_{\max}$ |
|---|---|---|---|---|
| LF-WikiSeeAlsoTitles-320K | 1024 | 300 | 0.0002 | 32 |
| LF-WikiTitles-500K | $\uparrow$ | $\uparrow$ | $\uparrow$ | $\uparrow$ |
| LF-WikiSeeAlso-320K | $\uparrow$ | $\uparrow$ | $\uparrow$ | 256 |
| LF-Wikipedia-500K | $\uparrow$ | $\uparrow$ | $\uparrow$ | $\uparrow$ |

## F.3. Label Quantile Creation

For Figure 2, labels were divided into 5 equi-voluminous quantiles. To each label $l \in [L]$, a popularity score $V_l = |\{i : y_{il} = +1\}|$ was assigned by counting number of training datapoints tagged with that label. The total volume of all labels was computed as $V_{\text{tot}} \stackrel{\text{def}}{=} \sum_{l \in [L]} V_l$. Labels were arranged in decreasing order of their popularity score $V_l$. Five label quantiles were then created so that the volume of labels in each bin, sum of popularity of labels, is roughly $\approx V_{\text{tot}}/5$. Thus, labels were collected in the first bin in decreasing order of popularity till the total volume of labels in that bin exceeded $V_{\text{tot}}/5$ at which point the first bin was complete and the second bin was created by selecting remaining labels in decreasing order or popularity till the total volume of labels in the second bin exceeded $V_{\text{tot}}/5$ and so on. For example, for the LF-WikiTitles-500K dataset, the five bins were found to contain approximately $1K, 9K, 30K, 84K, 375K$ labels respectively. Note that the first bin contains very few $\approx 1K$ labels since these are head labels and a small number of them quickly racked up a total volume of $\approx V_{\text{tot}}/5$ whereas the last quantile contains more than $100\times$ more labels at around $375K$ labels since these are tail labels and so a lot more of them are needed to add up to a total volume of $\approx V_{\text{tot}}/5$.

## F.4. Evaluation Metrics

Performance has been evaluated using propensity scored precision@$k$ and nDCG@$k$, which are unbiased and more suitable metric in the extreme multi-labels setting (Jain et al., 2016; Babbar & Schölkopf, 2019; Prabhu et al., 2018a;b). The propensity model and values available on The Extreme Classification Repository (Bhatia et al., 2016) were used. Performance has also been evaluated using vanilla precision@$k$ and nDCG@$k$ (with $k = 1, 3$ and 5) for extreme classification.

Let $\hat{\mathbf{y}} \in \mathbb{R}^L$ denote the predicted score vector and $\mathbf{y} \in \{0,1\}^L$ denote the ground truth vector (with $\{0,1\}$ entries this time instead of $\pm 1$ entries, for sake of convenience). The notation $rank_k(\hat{\mathbf{y}}) \subset [L]$ denotes the set of $k$ labels with highest scores in the prediction score vector $\hat{\mathbf{y}}$ and $\|\mathbf{y}\|_1$ denotes the number of relevant labels in the ground truth vector. Then we have:

$$\text{P@}k = \frac{1}{k} \sum_{l \in \text{rank}_k(\hat{\mathbf{y}})} y_l \tag{31}$$

$$\text{PSP@}k = \frac{1}{k} \sum_{l \in \text{rank}_k(\hat{\mathbf{y}})} \frac{y_l}{p_l} \tag{32}$$

$$\text{DCG@}k = \frac{1}{k} \sum_{\{(i,l)|l \in \text{rank}_k(\hat{\mathbf{y}}), i \in \mathbb{N}, i = |\{y \in \text{rank}_k(\hat{\mathbf{y}})|y<l\}|\}} \frac{y_l}{\log(i+1)} \tag{33}$$

$$\text{PSDCG@}k = \frac{1}{k} \sum_{\{(i,l)|l \in \text{rank}_k(\hat{\mathbf{y}}), i \in \mathbb{N}, i = |\{y \in \text{rank}_k(\hat{\mathbf{y}})|y<l\}|\}} \frac{y_l}{p_l \log(i+1)} \tag{34}$$

$$\text{nDCG@}k = \frac{\text{DCG@}k}{\sum_{l=1}^{\min(k,\|\mathbf{y}\|_0)} \frac{1}{\log(l+1)}} \tag{35}$$

$$\text{PSnDCG@}k = \frac{\text{PSDCG@}k}{\sum_{l=1}^{k} \frac{1}{\log l+1}} \tag{36}$$

$$\text{FN@}k = 1 - \frac{\sum_{l \in \text{rank}_k(\hat{\mathbf{y}})} y_l}{\|\mathbf{y}\|_1} \tag{37}$$

Here, $p_l$ is propensity score of the label $l$ calculated as described in Jain et al. (2016).

# G. Additional Experimental Results

We now present additional experimental results and various ablation experiments pertaining to MOGIC.

## G.1. Sensitivity Analysis of the Loss Hyperparameters

Results in Table 2 used $(\alpha, \beta) = (1, 0.1)$ which leads to the best performance. This section presents the ablations on the choice of $\alpha$ and $\beta$ on the LF-WikiSeeAlsoTitles-320K dataset, with the values of all other hyperparameters as defined in Appendix F.2. The following table summarizes the results. We also observe that the performance of MOGIC is generally robust to the choice of $(\alpha, \beta)$ with a variance of 0.108 in P@1, when comparing across all the choices considered. Figure 5 presents the results in the form of a heat map for better visualization.

Table 11: Sensitivity analysis over the $\alpha$ and $\beta$ hyperparameters, considering $\alpha \in \{0.1, 1, 10\}$ and $\beta \in \{0.1, 1, 10\}$

| $\alpha$ | $\beta$ | P@1 | P@5 | N@5 | PSP@1 | PSP@5 |
|---|---|---|---|---|---|---|
| 0.1 | 0.1 | 34.23 | 17.75 | 35.40 | 27.05 | 32.67 |
| 0.1 | 1.0 | 34.34 | 17.74 | 35.45 | 27.07 | 32.63 |
| 0.1 | 10.0 | 33.70 | 17.47 | 34.83 | 26.23 | 32.07 |
| **1.0** | **0.1** | **34.62** | **17.93** | **35.70** | **27.44** | **33.18** |
| *1.0* | *1.0* | *34.56* | *17.91* | *35.66* | *27.32* | *33.12* |
| 1.0 | 10.0 | 34.05 | 17.61 | 35.05 | 26.55 | 32.42 |
| 10.0 | 0.1 | 34.16 | 17.57 | 35.04 | 26.84 | 32.39 |
| 10.0 | 1.0 | 34.11 | 17.56 | 35.00 | 26.72 | 32.32 |
| 10.0 | 10.0 | 33.61 | 17.30 | 34.47 | 25.89 | 31.66 |

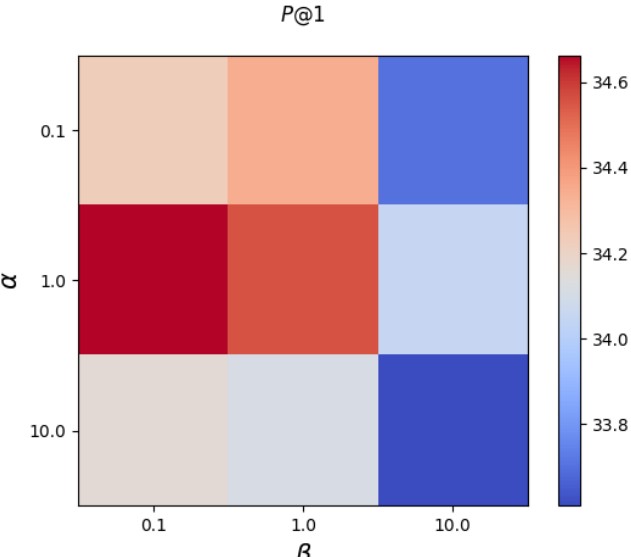

Figure 5: A plot of Table 11, which presents a sensitivity analysis over the $\alpha$ and $\beta$ hyperparameters, considering all combinations of $\alpha \in \{0.1, 1, 10\}$ and $\beta \in \{0.1, 1, 10\}$. Red denotes high values, whereas Blue denotes low values.

Table 12: The synergistic combination of the various components within MOGIC leads to the observed performance gains. This table shows ablations of MOGIC with different components on LF-WikiSeeAlsoTitles-320K dataset.

| Setting | Models | P@1 | P@5 | N@5 | PSP@1 | PSP@5 |
|---|---|---|---|---|---|---|
| 1 | MOGIC (OAK) | **34.62** | **17.93** | **35.70** | **27.44** | **33.18** |
| 2 | MOGIC (OAK) + Oracle w/o Metadata | *34.09* | *17.43* | *34.90* | *26.95* | *32.13* |
| 3 | Early Fusion (similar to REALM) | 28.49 | 14.52 | 29.46 | 22.26 | 26.52 |
| 4 | MOGIC (OAK) + Early Fusion | 29.30 | 14.88 | 29.92 | 22.21 | 26.87 |

**G.2. Contribution of Components to Performance Gain**

To systematically evaluate the individual contributions of the various components within the MOGIC framework, we conduct a series of ablation studies in this section (cf. Table 12).

**Contribution of metadata in oracle training (Setting 2 in Table 12):** We train the MOGIC (OAK) model using an oracle which has not been trained using any metadata. As seen in Table 12, this experiment yields results that are better than standard OAK but not as good as the proposed MOGIC (OAK) where the oracle has access to metadata during training. This shows that while individually, the regularization itself can result in better performance of the disciple, training oracle with access to metadata further improves the performance.

**Early fusion in disciple model (Settings 3 and 4 in Table 12):** We choose to perform a late fusion of the metadata information in the disciple model by passing the query representation and memory items through a single layer combiner. An alternate choice could be to perform early fusion of metadata. To validate this, we perform two experiments, (a) only early fusion in disciple (Setting 3), where the predicted metadata is concatenated with the input and (b) both early and late fusion (Setting 4) where alongside concatenating the predicted metadata at input (Table 12), we also perform the fusion using the combiner layer. Both of these approaches perform significantly worse than our proposed framework. This is because late fusion adds robustness to incorrect predictions in the metadata.

**G.3. Efficiency Analysis**

We now report the training and inference time and comparisons between oracle and MOGIC (OAK). Tables 13 and 14 summarize these numbers. As we already note in Section 4, MOGIC is merely a regularization framework, and we observe that inference times for a given baseline disciple model and its MOGIC variant are identical. There is marginal increase in the training time due to computation of the additional regularization terms in the loss function.

Table 13: Training and inference time of MOGIC (OAK) and DistilBERT oracle on different datasets.

| Dataset | Inference (in ms) | Training (in hrs) | Inference (in ms) | Training (in hrs) |
|---|---|---|---|---|
| | **MOGIC (OAK)** | | **DistilBERT Oracle** | |
| LF-WikiSeeAlsoTitles-320K | 14.06 | 25 | 24.70 | 45 |
| LF-WikiTitles-500K | 13.72 | 41 | 26.79 | 69 |
| LF-AmazonTitles-131K | 13.66 | 8 | 25.21 | 13 |

**Training**: All models were trained on AMD 4xMI200 GPUs. MOGIC (OAK) utilizes a context length of 32 for short-text datasets and 256 for full-text datasets. The Oracle model was trained with context lengths of 128 and 256 for short and full-text datasets, respectively. The training time for the DistilBERT-based oracle was approximately 2x that of MOGIC (OAK), primarily attributed to the increased context length. This longer context length results in larger attention matrices, thereby increasing the computational cost of backpropagation.

**Inference**: We evaluate the inference latency of MOGIC (OAK) on CPU using 8 threads, processing a single query with 2 metadata vectors within a PyTorch implementation. MOGIC (OAK) demonstrates significantly faster inference times compared to the oracle model.

Table 14: Real world deployment latency of MOGIC (OAK) and its OAK encoder in milliseconds.

| Algorithm | Mean | p95 | p99 | p999 |
|---|---|---|---|---|
| OAK encoder | 10.30 | 20.05 | 22.59 | 23.49 |
| MOGIC (OAK) | 14.95 | 22.34 | 25.07 | 27.08 |

To validate real-world feasibility, we further assess end-to-end production latency with inference on CPU, processing a single query with two metadata vectors. Through model optimizations, we significantly reduce the latency gap between the OAK encoder and MOGIC (OAK) inference, demonstrating the feasibility of deploying MOGIC (OAK) in real-world scenarios.

### G.4. Generalization of MOGIC on Other Datasets Across Disciples

MOGIC can generalize across both disciples and datasets. To validate this, we now include results on training MOGIC with the DEXA and NGAME disciples on LF-Amazon-131K dataset. Table 15 summarizes these results, wherein we observe that MOGIC demonstrates performance gains across both disciples.

Table 15: MOGIC is a general framework, and can be extended to any base XC algorithm to improve its accuracy. Along with LF-WikiSeeAlsoTitles-320K (Table 6), MOGIC also shows gains over other disciples on LF-AmazonTitles-131K. We observe MOGIC can improve accuracy of the base algorithm by 1-2% in P@1.

| Models | P@1 | P@5 | N@5 | PSP@1 | PSP@5 |
|---|---|---|---|---|---|
| MOGIC (OAK) | **47.01** | **22.40** | **49.51** | **40.62** | **50.33** |
| OAK | _46.42_ | _21.88_ | _49.06_ | _39.76_ | _49.78_ |
| MOGIC (NGAME) | **44.27** | **21.26** | **47.48** | **39.48** | **49.18** |
| NGAME | _43.44_ | _21.16_ | _47.10_ | _39.00_ | _49.00_ |
| MOGIC (DEXA) | **45.43** | **21.70** | **48.49** | **39.91** | **49.95** |
| DEXA | _44.47_ | _21.34_ | _47.65_ | _39.25_ | _49.08_ |

## G.5. Experimental Results on Full-text Datasets

Table 16: Results on full-text benchmark datasets demonstrate that MOGIC achieves up to 2% higher accuracy compared to baseline methods, similar to its performance on short-text datasets. Results on short-text datasets are presented in Table 2. For details on evaluation metrics, please refer to Appendix F.4.

| Method | P@1 | P@5 | N@5 | PSP@5 | P@1 | P@5 | N@5 | PSP@5 | P@1 | P@5 | N@5 | PSP@5 |
|---|---|---|---|---|---|---|---|---|---|---|---|---|
| | LF-WikiSeeAlso-320K | | | | LF-Wikipedia-500K | | | | LF-Amazon-131K | | | |
| MOGIC (OAK) | **49.62** | **24.26** | **50.49** | **43.17** | *85.34* | **51.50** | **77.85** | **61.74** | **50.05** | **23.72** | **52.87** | **53.80** |
| OAK | *48.57* | *23.28* | *49.16* | *40.44* | 85.23 | *50.79* | *77.26* | *60.80* | *48.36* | *22.20* | *51.27* | *52.21* |
| DEXA | 47.11 | 22.71 | 47.62 | 38.78 | 84.92 | 50.51 | 76.8 | 58.33 | 46.64 | 22.06 | - | 50.38 |
| NGAME | 46.4 | 18.05 | 46.64 | 33.33 | 84.01 | 49.97 | 75.97 | 57.04 | 46.53 | 22.02 | 49.58 | 50.45 |
| ANCE | 45.64 | 17.32 | 45.43 | 32.83 | 77.92 | 40.95 | 68.7 | 57.33 | - | - | - | - |
| DEXML | - | - | - | - | **85.78** | 50.53 | 77.11 | 58.97 | - | - | - | - |
| GraphFormers | 18.14 | 8.81 | 20.81 | 20.98 | 31.10 | 14.00 | 24.87 | 21.83 | - | - | - | - |
| GraphSAGE | 19.30 | 10.82 | 22.67 | 23.5 | 32.53 | 15.5 | 25.33 | 19.14 | - | - | - | - |

## G.6. Comparisons with Pure LLM-based Approaches

Table 17 compares MOGIC against LLaMA- and Phi-based oracles without disciple models, when LoRA-finetuned for label generation (XC task). Generative LMs cannot be directly employed for the XC task, and therefore, for these experiments we carry out classification by comparing the embeddings associated with final token of the query and the labels from the last layer of the SLM. Results reported in Table 17 summarizes this experiment. From Table 17, we observe that the (65M sized) MOGIC disciple, trained with a DistilBERT/LLaMA-based oracle outperforms the 7B-scale fine-tuned SLMs used directly without any disciple. We attribute this worse performance to the fact that these SLMs were pre-trained for text generation and then adapted for label generation. However, despite their performance, their embeddings, when used for regularization in the MOGIC framework, result in improved performance of the disciple models.

To further investigate performance comparisons against language models, we also include comparisons against LLaMA, Phi, and GPT when used directly for label generation, with and without LoRA fine-tuning.

Table 17: Comparison of DistilBERT, an embedding-based method, with SLMs trained for text generation.

| Model | Task Evaluation Method | P@1 | P@5 | PSP@1 | PSP@5 |
|---|---|---|---|---|---|
| DistilBERT (cf. Table 3) | Embedding based | **47.63** | **22.75** | **36.71** | **41.45** |
| LLaMA + Metadata (LoRA) (cf. Table 3) | Embedding based | *34.20* | *16.21* | *30.46* | *31.93* |
| Phi + Metadata (LoRA) (cf. Table 3) | Embedding based | 33.32 | 15.48 | 29.75 | 30.61 |
| LLaMA + Metadata (LoRA) | Generative | 9.427 | 9.065 | 12.86 | 9.982 |
| Phi + Metadata (LoRA) | Generative | 8.267 | 8.031 | 10.25 | 7.689 |
| GPT + Metadata | Generative | 14.57 | 12.26 | 16.86 | 12.92 |

## G.7. Dependency on Metadata During MOGIC Training

As discussed in Table 8 (cf. Section 4.3), disciples trained with an oracle are more robust to noisy/missing metadata in comparison to their oracle counterparts during test time. This robustness is not limited to noise at inference-time, but also noise present in the metadata during training time. To simulate a training scenario where reliable metadata linkages are not available, we consider the following setting: We train MOGIC wherein 50% of the metadata is corrupted (i.e., replaced with a randomly selected metadata from the corpus). The following table summarizes the linker, the oracle and disciple models' performance. We observe that MOGIC (OAK) is more robust to noisy metadata, even during training, and its drop performance is negligible. The MOGIC model trained with noisy metadata continues to be state of the art, outperforming the other baselines.

Table 18: Linker metrics with 50% noise-added metadata, where 50% metadata for each query is replaced by random metadata.

| Method | P@1 | P@5 | N@5 | PSP@1 | PSP@5 | P@1 | P@5 | N@5 | PSP@1 | PSP@5 |
|---|---|---|---|---|---|---|---|---|---|---|
| | LF-WikiSeeAlsoTitles-320K | | | | | LF-WikiTitles-500K | | | | |
| Ground-truth metadata | **46.16** | **20.97** | **36.48** | **28.75** | **25.37** | **20.74** | **13.72** | **15.90** | **7.38** | **8.29** |
| Noisy metadata | *27.98* | *9.90* | *20.82* | *21.62* | *14.10* | *10.06* | *4.99* | *6.44* | *7.12* | *5.48* |

Table 19: Oracle metrics with 50%-noisy metadata

| Method | P@1 | P@5 | N@5 | PSP@1 | PSP@5 | P@1 | P@5 | N@5 | PSP@1 | PSP@5 |
|---|---|---|---|---|---|---|---|---|---|---|
| | LF-WikiSeeAlsoTitles-320K | | | | | LF-WikiTitles-500K | | | | |
| Ground-truth Oracle | **47.48** | **22.64** | **48.18** | **36.60** | **41.27** | **64.32** | **29.92** | **50.58** | **37.41** | **39.75** |
| Noisy Oracle | *41.23* | *19.92* | *42.17* | *30.94* | *35.77* | *59.85* | *26.98* | *46.47* | *34.58* | *36.00* |

Table 20: Performance of MOGIC (OAK) with 50%-noisy metadata during training

| Method | P@1 | P@5 | N@5 | PSP@1 | PSP@5 | P@1 | P@5 | N@5 | PSP@1 | PSP@5 |
|---|---|---|---|---|---|---|---|---|---|---|
| | LF-WikiSeeAlsoTitles-320K | | | | | LF-WikiTitles-500K | | | | |
| MOGIC (OAK) (cf. Table 2) | **34.62** | **17.93** | **35.70** | 27.44 | **33.18** | **47.28** | **18.55** | **34.97** | 27.29 | 26.12 |
| MOGIC (OAK) with 50%-noisy metadata | *34.29* | *17.68* | *35.37* | *27.89* | *33.07* | *46.68* | *18.53* | *34.83* | *27.44* | *26.27* |

## G.8. Experimental Results on LF-AmazonTitles-1.3M

We present the results of MOGIC (OAK) on the LF-AmazonTitles-1.3M dataset, which is an extremely large-scale extreme classification (XC) benchmark containing approximately 1.3 million labels.

Table 21: Results on LF-AmazonTitles-1.3M benchmark datasets

| Method | Metadata | P@1 | P@5 | N@5 | PSP@1 | PSP@3 | PSP@5 |
|---|---|---|---|---|---|---|---|
| MOGIC (OAK) | Dump Category | 48.93 | 38.49 | 46.45 | *35.78* | *38.92* | *40.59* |
| OAK | Dump Category | 48.91 | 38.13 | 46.07 | 34.65 | 37.53 | 39.04 |
| MOGIC (OAK) | GPT Category | **50.95** | **39.95** | **48.19** | **36.28** | **39.50** | **41.22** |
| OAK | GPT Category | *49.46* | *38.61* | *46.62* | 34.92 | 37.86 | 39.41 |

We constructed metadata for the LF-AmazonTitles-1.3M dataset using two different approaches:

1. **Product-dump parsing**: We parsed the Amazon product data dump to extract auxiliary information related to each product. Among the various fields available, we selected the product category as the metadata to be used in our experiments.

2. **GPT-generated categories**: We used GPT-4o-mini to generate category labels for each product, which were then used as metadata.

Empirically, we observed that the GPT-generated categories yielded the most significant performance improvements. However, the generated categories were not always consistent. To ensure compatibility with our algorithm, we performed a conflation step to normalize and standardize these categories before use. For the conflation, we identified all tail entities (having fewer than 5 samples per entity) and replaced these labels with their closest semantic match from the entire entity set. A threshold was applied before the conflation to ensure that no two unrelated categories are merged together. This was done to ensure that the erroneous entities generated due to stochastic nature of language models are merged together with a more general entity label.

# H. Case Study: Effect of Metadata in MOGIC (OAK)

In our framework, metadata is provided as early concatenation in textual form to the oracle, which is then used to regularize the free parameters of the disciple's memory during training. While OAK suffers from overgeneralization and semantic drift—often selecting popular or tangentially related labels—MOGIC (OAK) leverages metadata to guide its predictions toward domain-specific and contextually appropriate labels. Metadata serves three main functions:

- **Disambiguation**: helping resolve terms with multiple meanings.

- **Semantic grounding**: encouraging consistency in domain or region.

- **Error pruning**: reducing the model's tendency to retrieve globally frequent but contextually irrelevant entities.

Let us consider the two queries presented in Table 1 taken from LF-WikiSeeAlsoTitles-320K, where the task is, given the title of a Wikipedia article, predict related Wikipedia articles listed in its *See Also* section. In the following, we examine how metadata helps MOGIC rectify errors from OAK.

1. **Grass court**

   For the query "Grass court", the ground truth labels include specific types of tennis courts (Clay court, Carpet court, Hardcourt), which are all surface types in the sport of tennis. The predicted metadata included Courts by type, Landforms and Grasslands. Out of which Courts by type is the only relevant metadata which links to the ground-truth labels.

   OAK predictions are completely off-topic, retrieving labels such as Fernie Ghostriders (a Canadian ice hockey team), Garland, Texas (a geographic location), and Ronald Reagan Boyhood Home (a historical site). These results reflect a semantic drift: the model picks up on vague or shallow lexical or associative signals related to "court" but fails to situate it in the correct sports domain.

   MOGIC (OAK), in contrast, correctly predicts all three ground truth labels and only makes minor errors (e.g., Video arcade, U.S. Men's Clay Court Championships). This improvement comes from the regularization-based training of MOGIC which helps the disciple focus on the right metadata using the guidance received from the oracle model during training.

   This example highlights how metadata helps disambiguate multi-sense terms (e.g., "court") and offers semantic grounding by restricting the model's attention to the correct conceptual subspace.

2. **Tangbe**

   The query "Tangbe" refers to a village in the Mustang District of Nepal. The ground truth labels reflect this geographical and cultural context: Mustang District, Kali Gandaki Gorge, Upper Mustang, etc. The predicted metadata includes Populated places in Cameroon, Communes of Cameroon and Township-level divisions of Hebei which are all irrelevant.

   OAK predictions include entities like Desalpur (an archaeological site in India), Vladivostok (a city in Russia), and Kitenge (a textile), which are geographically and topically irrelevant. The model appears to be guessing from a broad set of vaguely place-related or historical entities without any clear regional grounding, which is mostly likely due to error propagation from the linker.

   In contrast, MOGIC (OAK) correctly retrieves all five ground truth labels, suggesting a precise understanding of the regional context. This highlights the error pruning capability of MOGIC.

