# OpenReview forum: "MOGIC: Metadata-infused Oracle Guidance for Improved Extreme Classification"
_ICML.cc/2025/Conference — ICML 2025 poster_

### Official Review · Reviewer_3zWG · 2025-03-12

**Overall Recommendation:** 4

**Summary:**

This paper mainly explores methods to enhance classification performance using metadata in the task of Extreme Classification. Experiments on six popular benchmark datasets show that the method significantly improves the model performance.

**Claims And Evidence:**

Yes

**Essential References Not Discussed:**

No

**Experimental Designs Or Analyses:**

Yes

**Methods And Evaluation Criteria:**

Yes

**Other Comments Or Suggestions:**

See weakness above

**Other Strengths And Weaknesses:**

1. The proposed MOGIC framework is innovative as it combines early-fusion of text-based metadata and late-fusion of memory items.
2. The two-phase training process involving Oracle training and Oracle-guided disciple training might introduce additional complexity.
3. Although it shows good performance on the tested datasets, its scalability to extremely large-scale or rapidly evolving datasets remains to be seen.

**Questions For Authors:**

See weakness above

**Relation To Broader Scientific Literature:**

This paper makes solid contributions to the area of Extreme Classification.

**Theoretical Claims:**

Yes, in Theoretical Justification of Oracle-Guided Losses

---

> ### Author Rebuttal · Authors · 2025-04-01
>
> We thank the reviewer for their detailed review. Please find our response to your comments below.
>
> 1. **The two-phase training process involving Oracle training and Oracle-guided disciple training might introduce additional complexity.**
> * **Response**: We agree with the reviewer that the two-stage training framework introduces some additional complexity compared to single-stage baselines. However, the improvements in performance and the increased robustness achieved through regularization in this two-stage approach present a reasonable tradeoff for the added complexity. Moreover, this complexity is limited to the training phase (a one-time cost), while the deployed disciple models remain highly efficient at inference.
>
> 2. **Although it shows good performance on the tested datasets, its scalability to extremely large-scale or rapidly evolving datasets remains to be seen.**
>
> * **Response**: The MOGIC approach can be readily scaled to larger datasets. However, there are no high-quality rapidly-evolving dataset for XC, to the best of our understanding. We believe this is a strong area for future research. The Wikipedia-500k is one of the largest datasets used by the XC community, comprising 1.8M training samples and 501K labels, while many other XC datasets such as EURLex-4.3K, Bibtex or AmazonCat-13K have significantly fewer labels (such as 4.3K in the EURLex-4.3K dataset or 13K in the AmazonCat-13K dataset). While LF-AmazonTitles-1.3M is another viable choice, we reported results on Amazon-131k, since metadata in this setting is not readily available, and must be generated using GPT-based approaches. In light of your comments, we are also working to include experimentation on the LF-AmazonTitles-1.3M dataset. While we will include these results in the final version of the manuscript, given the additional resources involved in generating the metadata and subsequently training the models, we do not have the results ready to report at this time, but are striving to incorporate them before the end of the rebuttal discussion phase.

---

### Official Review · Reviewer_FoCK · 2025-03-13

**Overall Recommendation:** 3

**Summary:**

The paper introduces MOGIC, a framework for improving extreme classification (XC) by leveraging metadata through a two-phase training approach. XC involves tasks with extremely large label spaces (e.g., product recommendations, Wikipedia tagging) where metadata can enhance accuracy but faces challenges like noise and latency. Existing methods use late-stage fusion for efficiency but underperform when metadata is clean. MOGIC trains an early-fusion Oracle model with access to ground-truth metadata to guide a disciple model (e.g., OAK, NGAME) via regularization. This approach improves precision and robustness without increasing inference latency, achieving 1–2% gains on six datasets.

## update after rebuttal
Thanks for the responses and extra sensitivity analysis which partially addresses my questions. I will keep my positive rating of 3 considering the overall quality of this paper.

**Claims And Evidence:**

Yes

**Essential References Not Discussed:**

Nil.

**Ethical Review Concerns:**

Nil.

**Experimental Designs Or Analyses:**

Six datasets are used for the experiments. Three different LLMs are tested and many baselines are compared. I think the experimental evaluation is quite comprehensive.

**Methods And Evaluation Criteria:**

1. The method leverages both query-side and label-side metadata, enriching representations bidirectionally. Examples show improved label prediction by incorporating contextual metadata from both ends.
2. MOGIC demonstrates consistent performance gains across multiple datasets and metrics. It improves precision@1, NDCG, and propensity-scored metrics over state-of-the-art models like OAK. The method is validated on six benchmarks, showing broad applicability.

**Other Comments Or Suggestions:**

Comparisons with pure LLM-based approaches (without disciple models) are missing. Larger Oracles might outperform MOGIC if computational constraints are relaxed, but this trade-off is not quantified.

**Other Strengths And Weaknesses:**

1. The approach is model-agnostic, enhancing both memory-based (OAK) and memory-free (NGAME, DEXA) XC models. Plug-and-play compatibility allows integration into existing architectures. Flexibility in Oracle choice (DistilBERT, Phi-2) balances performance and efficiency.
2. Theoretical analysis justifies the regularization losses, linking disciple performance to Oracle-guided training. Bounds on population loss show the disciple converges toward Oracle accuracy with finite samples.
3. MOGIC maintains low inference latency by avoiding early-fusion overhead. Training costs are manageable, and inference matches base models’ speed. Experiments confirm no latency increase compared to OAK.
4. Robustness to missing or noisy metadata is demonstrated through quantile-wise analysis and noise injection tests. The disciple outperforms the Oracle when metadata is perturbed, showing resilience to real-world conditions.

**Questions For Authors:**

1. The Oracle’s dependency on high-quality metadata during training limits performance if metadata is sparse or biased. While MOGIC handles inference-time noise, training assumes reliable metadata linkages, which may not hold in all scenarios.
2. Hyperparameters α and β for loss balancing are set empirically. The paper does not analyze sensitivity to these choices, risking suboptimal tuning in new applications.
3. The framework may overfit Oracle biases, particularly if the Oracle’s metadata integration is noisy. Ablation studies show performance drops without metadata, hinting at potential over-reliance.

**Relation To Broader Scientific Literature:**

This work is related to extreme classification. And it is also related to RAG in LLMs.

**Theoretical Claims:**

There are some theoretical proofs which seem to be correct. One typo is that, the symbol k in Inequality (11) should be K.

---

> ### Author Rebuttal · Authors · 2025-04-01
>
> Thank you for your detailed review. Below are our responses to your comments.
>
> 1. **Comparisons with LLM-based approaches**
> * **Response**: We have already included comparisons of MOGIC against LLaMA and Phi-based Oracles without disciple models, when LoRA-finetuned for label generation (XC task) in Tab 3 (and row 1-3 below). Generative LMs cannot be directly employed for the XC task, and therefore, for these experiments we carry out classification by comparing the embeddings associated with final token of the query and the labels from the last layer of the SLM. Results reported in Tab. 3 of the paper summarizes this experiment (cf. Lines L367-380, col 1).  We observe that the (65M sized) MOGIC disciple, trained with a DistilBERT/LLaMA-based oracle outperforms the 7B-scale fine-tuned SLMs used directly without any disciple. We attribute this worse performance to the fact that these SLMs were pre-trained for text generation and then adapted for label generation. However despite their performance, their embeddings, when used for regularization in the MOGIC framework, result in improved performance of the disciple models. We will improve the clarity of this discussion.
>
> *  To further investigate performance comparisons against language models, as part of the rebuttal, we also include comparisons against LLaMA, Phi, and GPT when used directly for label generation (row 4-6). We observe that oracle models trained specifically for the XC task perform better than much larger LMs. Also, for the XC task, using the embedding from LMs is better than mapping their generations to the label set since the generations often contain labels which do not belong to the label set.
>
> *Oracle metrics on LF-WikiSeeAlsoTitles-320K*
> ||P@1|P@5|
> |-|-|-|
> |DistilBERT (65M) (cf.Table3)|47.63|22.75|
> |LLaMA-2 (7B) (LoRA) -Embed (cf.Table3)|34.20|16.21|
> |Phi-2 (2.7B) (LoRA) -Embed (cf.Table3)|33.32|15.48|
> |LLaMA-2 (7B) (LoRA) Gen|9.427|9.065|
> |Phi-2 (2.7B) (LoRA) Gen|8.267|8.031|
> |GPT 4-o Gen|14.57|12.26|
>
> 2. **Metadata dependency during training**
> * **Response**: As discussed in Tab. 8 (cf. Sec 4.3), disciples trained with an Oracle are more robust to noisy/missing metadata in comparison to their Oracle counterparts during inference. This robustness is not limited to noise at inference, but also noise present in the metadata during training. To simulate a training scenario wherein reliable metadata linkages are not available, we consider the following setting: We train MOGIC on LF-WikiSeeAlsoTitles-320K dataset wherein 50% of the metadata is replaced with a randomly selected metadata from the corpus. The following table summarizes the linker, the Oracle and disciple models' performance. We observe that MOGIC (OAK) is more robust to noisy metadata, even during training with negligible drop in performance. The MOGIC model trained with noisy metadata, continues to be SOTA, outperforming other baselines.
>
> *Linker metrics with 50% noise-added metadata, where 50 % metadata for each query is replaced by random metadata.*
> ||P@1|P@5|N@5|PSP@1|PSP@5|
> |-|-|-|-|-|-|
> |Ground-truth metadata|46.16|20.97|36.48|28.75|25.37|
> |Noisy metadata|27.98|9.90|20.82|21.62|14.10|
>
> *Oracle metrics with 50%noisy metadata*
> ||P@1|P@5|N@5|PSP@1|PSP@5|
> |-|-|-|-|-|-|
> |Ground truth Oracle|47.48|22.64|48.18|36.60|41.27|
> |Noisy Oracle|41.23|19.92|42.17|30.94|35.77|
>
> *Performance of MOGIC (OAK) and OAK with 50%noisy metadata during training*
> ||P@1|P@5|N@5|PSP@1|PSP@5|
> |-|-|-|-|-|-|
> |MOGIC(OAK) (cf.Table2)|34.62|17.93|35.70|27.44|33.18|
> |MOGIC(OAK) with 50%noisy metadata|34.29|17.68|35.37|27.89|33.07|
>
> 3. **Sensitivity of Loss Balancing Hyperparameters**
> * **Response**: We now include a sensitivity analysis over the $𝛼$ and $β$ hyperparameters, considering all combinations of $𝛼\in\{0.1,1,10\}$ and $β\in\{0.1,1,10\}$. We performed a sensitivity analysis prior to our experimentation and found that $(𝛼,β) = (1,0.1)$ leads to the best performance, which are the values we use for the experiments in the paper. We now present the ablations on the choice of $𝛼$ and $β$ on the LF-WikiSeeAlsoTitles-320K dataset, with all other hyperparameters as defined in Appendix H. This table summarizes the results. We observe that $(𝛼,β) = (1,0.1)$ results in the best performance. However, the performance of MOGIC is generally robust to the choice of $(𝛼,β)$ with a variance of 0.108 in P@1, when comparing across all the choices considered. We will improve the clarity of this hyperparameter choice in the final version of the manuscript, and include this ablation in the Appendix.
>
> |$𝛼$|$β$|P@1|P@5|N@5|PSP@1|PSP@5|
> |-|-|-|-|-|-|-|
> |0.1|0.1|34.23|17.75|35.40|27.05|32.67|
> |0.1|1.0|34.34|17.74|35.45|27.07|32.63|
> |0.1|10.0|33.70|17.47|34.83|26.23|32.07|
> |1.0|0.1|34.62|17.93|35.70|27.44|33.18|
> |1.0|1.0|34.56|17.91|35.66|27.32|33.12|
> |1.0|10.0|34.05|17.61|35.05|26.55|32.42|
> |10.0|0.1|34.16|17.57|35.04|26.84|32.39|
> |10.0|1.0|34.11|17.56|35.00|26.72|32.32|
> |10.0|10.0|33.61|17.30|34.47|25.89|31.66|

---

> > ### Comment · Reviewer_FoCK · 2025-04-03
> >
> > Thanks for the responses and extra sensitivity analysis which partially addresses my questions. I will remain my rating at 3 considering the overall quality of this paper.

---

> > > ### Author Response · Authors · 2025-04-07
> > >
> > > Dear Reviewer
> > >
> > > We thank you for your suggestions. We would like to present some more results to further address your questions.
> > >
> > >
> > > - **Sensitivity of Loss Balancing Hyperparameters**
> > >
> > > To strengthen the sentivity analysis of MOGIC w.r.t to hyperparameters, we have also added the numbers of LF-WikiTitles-500K dataset. Since this is a large dataset and requires extensive amount of compute, for the rebuttal, we report numbers after 50 epochs of training on the three best performing $\alpha$, $\beta$  combinations from LF-WikiSeeAlsoTitles-320K dataset.  The observation is consistent with that from the previous analysis and the $\alpha$, $\beta$ combination of 1.0, 0.1 performs the best. The performance of MOGIC is generally robust to hyperparameters, and the choice of hyperparameters seems to be agnostic of the dataset.
> > >
> > >
> > > * *Hyperparameter sensitivity analysis on LF-WikiTitles-500K*
> > >
> > > |   |  α  |  β   |   P@1 |   P@5 |   N@5 | PSP@1 | PSP@5 |
> > > |--:|-----|------|------:|------:|------:|------:|------:|
> > > | 1 | 1.0 |  0.1 | 43.92 | 17.04 | 32.57 | 27.09 | 24.71 |
> > > | 2 | 1.0 |  1.0 | 42.87 | 16.75 | 32.07 | 27.00 | 24.56 |
> > > | 3 | 1.0 | 10.0 | 43.42 | 16.67 | 31.98 | 26.49 | 24.14 |
> > >
> > >
> > > - **Metadata Dependency During Training and Oracle Bias**
> > >
> > > To further show the extent of the robustness of proposed framework to unreliable metadata during training, we have also extended the previous experiments to now include the  LF-WikiTitles-500K  dataset along with the LF-WikiSeeAlsoTitles-320K dataset. These results further go on to prove that MOGIC (OAK) is robust to noisy metadata, even during training with negligible drop in performance. This also proves that Oracle models trained with noisy data can still be used to train the disciple model.
> > >
> > > * *Linker metrics with 50% noise-added metadata, where 50 % metadata for each query is replaced by random metadata*
> > >
> > > |                       |   P@1 |   P@5 |   N@5 | PSP@1 | PSP@5 |
> > > |----------------------:|------:|------:|------:|------:|------:|
> > > | Ground-truth metadata | 20.74 | 13.72 | 15.90 |  7.38 |  8.29 |
> > > |        Noisy metadata | 10.06 |  4.99 |  6.44 |  7.12 |  5.48 |
> > >
> > >
> > > * *Oracle metrics with 50% noisy metadata*
> > >
> > > |                       |   P@1 |   P@5 |   N@5 | PSP@1 | PSP@5 |
> > > |----------------------:|------:|------:|------:|------:|------:|
> > > | Ground-truth metadata | 64.32 | 29.92 | 50.58 | 37.41 | 39.75 |
> > > |        Noisy metadata | 59.85 | 26.98 | 46.47 | 34.58 | 36.00 |
> > >
> > > * *Performance of MOGIC (OAK) with 50%-noisy metadata during training*
> > >
> > > |                                    |   P@1 |   P@5 |   N@5 | PSP@1 | PSP@5 |
> > > |-----------------------------------:|------:|------:|------:|------:|------:|
> > > | MOGIC(OAK) (cf.Table2) | 47.28 | 18.55 | 34.97 | 27.29 | 26.12 |
> > > |  MOGIC(OAK) with 50% noisy metadata | 46.68 | 18.53 | 34.83 | 27.44 | 26.27 |
> > >
> > > - **Comparisons with LLM-based approaches**
> > >
> > > We have also added the propensity metrics for results from generative language models. The same trends follow.
> > >
> > > |            |   P@1 |   P@5 |  PSP@1 | PSP@5 |
> > > |-----------:|------:|------:|------:|------:|
> > > | DistilBERT (cf. Table 3)  | 47.63  | 22.75 | 36.71 | 41.45 |
> > > | LLaMA+Metadata (LoRA) - Embed (cf. Table 3) | 34.20 | 16.21 |  30.46 | 31.93 |
> > > | Phi+Metadata (LoRA) - Embed (cf. Table 3) | 33.32 | 15.48 |  29.75 | 30.61 |
> > > | LLaMA+Metadata (LoRA) Gen | 9.427 | 9.065  | 12.86 | 9.982 |
> > > | Phi+Metadata (LoRA) Gen | 8.267 | 8.031 | 10.25 | 7.689 |
> > > | GPT+Metadata  Gen | 14.57 | 12.26 | 16.86 | 12.92 |
> > >
> > > - **Testing on Large-Scale XML Datasets**
> > >
> > > As suggested by other reviewers, we have shown results of MOGIC(OAK) on LF-AmazonTitles-1.3M dataset which is an extremely large scale XML dataset with 1.3 million labels. We observe that our framework MOGIC(OAK) shows gains over OAK.
> > >
> > > * *Results on LF-AmazonTitles-1.3M benchmark datasets*
> > >
> > > |             |   P@1 |   P@5 |   N@5 | PSP@1 | PSP@3 | PSP@5 |
> > > |------------:|------:|------:|------:|------:|------:|------:|
> > > | MOGIC (OAK) | 48.93 | 38.49 | 46.45 | 35.78 | 38.92 | 40.59 |
> > > |         OAK | 48.91 | 38.13 | 46.07 | 34.65 | 37.53 | 39.04 |
> > >
> > >
> > > ---
> > > ---
> > >
> > > We sincerely thank the reviewers for their constructive feedback, which has significantly improved the quality of our work. We warmly welcome any additional suggestions for further enhancement.

---

### Official Review · Reviewer_5PFN · 2025-03-14

**Overall Recommendation:** 4

**Summary:**

The authors propose a framework for building a disciple model which can perform extreme multi-label classification with the assistance of RAG-like metadata. This pipeline, MOGIC, is two-phase: in phase (1) an oracle with access to high-quality, ground-truth metadata is trained. In phase (2), a smaller, "disciple" model participates in knowledge "distillation-like" training to mimic the predictions of the oracle model but also make predictions on relevant metadata which might help with the downstream prediction. Through MOGIC, the disciple model is both robust to noisy metadata and performant on XML tasks such as WikiTitles-500K and AmazonTitles-131K.

## Update after rebuttal

I am choosing to maintain my positive score in light of the rebuttal.

**Claims And Evidence:**

The results, especially the P@1 numbers, indicate the effectiveness of the approach. It is to my understanding that through this strategy, lightweight architectures such as DistilBERT can perform on par with Phi-2 and Llama-2. It is clear through the experiments section that usage of high-quality textual metadata is capable of training a high-quality disciple model.

**Essential References Not Discussed:**

N/A

**Experimental Designs Or Analyses:**

The experimental designs appear to be fair and routine. The datasets are standard and frequently-encountered in XML literature. The augmentation of the XML samples and labels with metadata is new to me, but the construction of the metadata makes sense.

**Methods And Evaluation Criteria:**

I do believe that the proposed method makes sense for the problem, especially in the age of LLMs. Classical XML architectures and papers from 2016-2024 did not extensively consider the addition of metadata for help with classification, but given the typically lackluster results on popular XML datasets (due to their extreme difficulty), calling RAG-like methodology is intuitive. The datasets are appropriate, though I would have liked to have seen performance on even more challenging datasets such as Amazon-670k (see questions below).

**Other Comments Or Suggestions:**

The authors could try to present a simple example of the features and labels for an XML problem. It likely won't be immediately clear to a reader outside of this field that the label is an enormously-sized, highly sparse binary vector corresponding to classes -- this is only briefly covered. The authors should also try to further emphasize hat these problems are difficult because there are very few samples per label.

**Other Strengths And Weaknesses:**

Other Strengths:
-I find the contribution useful as it re-frames the XML problem within the landscape of LLMs and RAG. It's a fresh perspective.

Other Weaknesses:
-The disciple training doesn't seem entirely novel. It resembles knowledge-distillation, except it's not immediately clear if the disciple architecture is smaller or simply the same as the oracle.

**Questions For Authors:**

Is it possible to run tests on extremely challenging datasets such as WikiLSHTC-325K and Amazon-670K? I would be interested in seeing how metadata is constructed for these datasets and how MOGIC holds up. If not, could the authors explain why these datasets are not suitable for the described framework? I am also interested in seeing if a gold-standard method such as SLEEC can be defeated via MOGIC.

**Relation To Broader Scientific Literature:**

Extreme multi-label classification is applicable to the broader recommender systems community. In my opinion, XML is still generally unsolved and challenging, so the contribution of new algorithms which pushes the P@1 on these datasets is meaningful and useful to the broader machine learning community.

**Theoretical Claims:**

I did not check the correctness of the proof, though the bound makes sense as it becomes tighter with more metadata and samples. The authors should probably explain what the Rademacher constants are for those readers less familiar with PAC theory (are these just Rademacher splitting dimensions?).
.

---

> ### Author Rebuttal · Authors · 2025-04-01
>
> Thank you for your detailed review. Please find our response to your comments below.
>
> 1. **Clarity on Rademacher constants**
> * **Response**: The Rademacher complexity constants $R_q$ and $R_l$ in Theorem 1 are scalar values which quantify the complexity or capacity of the hypothesis classes corresponding to the query tower and label tower, respectively. We use the standard definitions of Rademacher constants from the Statistical Learning Theory for Binary Classification problems (ref: https://www.cs.cmu.edu/~ninamf/ML11/lect1117.pdf).
> * *Explanation on Rademacher complexity:* Mathematically, Rademacher complexity constants $R_q,R_l$ are estimated as the average empirical loss of minimizing the hypothesis class on a data sample with randomly annotated labels i.e. labels are generated by a purely random Bernoulli distribution with probabilities 0.5 to be either positive or negative. In intuitive terms, the smaller the values of $R_q, R_l$, the less prone are the query tower and label tower to overfit the finite training data, and consequently the accuracy on the test set is expected to be better.
>
> 2. **Novelty of  disciple training and clarity of oracle architecture**
> * **Response**: The disciple training in MOGIC is a novel variant of knowledge distillation (KD). Compared to standard KD, it differs in multiple ways. First, the metadata is provided as early concatenation in textual form for the oracle, but is used to train free parameters in the disciple's memory via a novel regularization framework. This can be viewed as KD from an early-concatenation model to a two-tower model. While all disciple models benefit from this framework, the additional parameters present in the memory-based disciples such as MOGIC (OAK) demonstrate the most gains. Furthermore, the oracle has access to ground-truth metadata, which is privileged information that is not available to the disciple model (unlike standard KD). In this MOGIC framework, the oracle can either be larger, or of the same size as the disciple. Table 3 of the main manuscript presents comparisons between larger (LLM-based, 2.7B/7B sized) oracle and a disciple-sized (65M sized), DistilBERT oracle (cf. Lines 367-380, column 1).
>
> 3. **Clarifying XML Labels and Data Scarcity**
>
> * **Response**: We will update the introduction of the manuscript accordingly. We will also include a detailed example, as suggested by you, in the appendix in the revised draft.
>
> 4. **Testing on Large-Scale XML Datasets like WikiLSHTC-325K or Amazon-670K**
>
> * **Response**: Datasets such as WikiLSHTC-325K (which contains 325K labels) do not contain raw text or label features associated with labels, and therefore have not been directly used for training encoder-based models such as NGAME, DEXA, DEXML, ANCE etc. This is why we chose to report on LF-WikiTitles-500k and LF-Wikipedia-500K dataset (which has 501K labels), which are from the same distribution and have the same task as WikiLSHTC (category prediction), but contain a larger label set, with label features/text available. Similarly, to be aligned with the baseline methods, we use LF-Amazon-131k dataset instead of Amazon-670K. To demonstrate MOGIC's effectiveness on larger-scale datasets, we are working to include experiments on the LF-AmazonTitles-1.3M dataset in the final version of the manuscript. While the results are not yet available due to the resource-intensive process of generating metadata and training models, we aim to include them before the conclusion of the rebuttal phase.
>
> 5. **Comparison against SLEEC.**
>
> * **Response**: SLEEC does not use label-text for classification, and therefore performs poorer than methods such as Parabel [1] on multiple standard datasets. A direct comparison against SLEEC is challenging, as we were unable to find an up-to-date implementation of the algorithm. Therefore, we choose to compare MOGIC against Parabel, which has been shown to outperform SLEEC across datasets[1]. We present these comparisons on LF-AmazonTitles-131K, LF-WikiSeeAlsoTitles-320K and LF-WikiTitles-500K. These results are reported in the table below, with performance numbers for Parabel obtained from ECLARE[2]:
>
> * *Performance of MOGIC(OAK) and Parabel on different benchmark datasets*
>
> |	|   P@1 |   P@5  | PSP@1 | PSP@5 |
> |--:|--:|--:|--:|--:|
> |  **LF-AmazonTitles-131K** |   |	|	|   |
> | MOGIC(OAK)  | 47.01  | 22.40  | 40.62 | 50.33 |
> | Parabel | 32.6 | 15.61 | 23.27 | 32.14 |
> |  **LF-WikiSeeAlsoTitles-320K** |   |	|	|   |
> | MOGIC (OAK) | 34.62 | 17.93 | 27.44 | 33.18 |
> | Parabel | 17.68 |  8.59 | 9.24 | 11.8 |
> |  **LF-WikiTitles-320K** |   |	|	|   |
> | MOGIC (OAK) | 47.28 | 18.55 | 27.29 | 26.12 |
> | Parabel | 40.41 |  15.42 | 15.55 |  15.35 |
>
>
> [1] Prabhu, Yashoteja, et al. "Parabel: Partitioned label trees for extreme classification with application to dynamic search advertising." World Wide Web Conference 2018.
>
> [2] Mittal, Anshul, et al. "ECLARE: Extreme classification with label graph correlations." theWebConf  2021.

---

### Decision · Program_Chairs · 2025-05-01

**Decision:**

Accept (poster)

**Comment:**

The paper introduces MOGIC, a method for improving Extreme Classification (XC). It is trains an "teacher" model with privileged, high-quality metadata and then distills into a student XC model with regularization. Experiments were conducted on standard XC benchmarks such as LF-AmazonTitles and LF-WikiTitles and showed performance gains in their setting without increasing inference latency. All the reviewers were positive about the paper and still raised questions about comparisons with purely LLM-based methods, robustness to unreliable metadata during training, hyperparameter sensitivity, and the method's performance on larger-scale datasets. The authors provided additional results showing robustness and addressing reviewers' initial concerns about scalability and practicality. Reviewers acknowledged the responses positively, maintaining their support for the paper based on clarified and supplemented points. Please incorporate these new results and reviewer feedback in the final version.